# Genetic variants associated with psychiatric disorders are enriched at epigenetically active sites in lymphoid cells

Mary-Ellen Lynall [1,2,3,4] ✉, Blagoje Soskic [5,6,7], James Hayhurst [6], Jeremy Schwartzentruber [5], Daniel F. Levey [8,9], Gita A. Pathak [8,9], Renato Polimanti [8,9], Joel Gelernter [8,9,10], Murray B. Stein [11,12], Gosia Trynka [5,6], Menna R. Clatworthy [3,4] & Ed Bullmore [1,2]

Multiple psychiatric disorders have been associated with abnormalities in both the innate and adaptive immune systems. The role of these abnormalities in pathogenesis, and whether they are driven by psychiatric risk variants, remains unclear. We test for enrichment of GWAS variants associated with multiple psychiatric disorders (cross-disorder or trans-diagnostic risk), or 5 specific disorders (cis-diagnostic risk), in regulatory elements in immune cells. We use three independent epigenetic datasets representing multiple organ systems and immune cell subsets. Trans-diagnostic and cis-diagnostic risk variants (for schizophrenia and depression) are enriched at epigenetically active sites in brain tissues and in lymphoid cells, especially stimulated CD4+ T cells. There is no evidence for enrichment of either trans-risk or cis-risk variants for schizophrenia or depression in myeloid cells. This suggests a possible model where environmental stimuli activate T cells to unmask the effects of psychiatric risk variants, contributing to the pathogenesis of mental health disorders.

Diagnostic systems for mental health disorders are comprised of multiple, categorically distinct clinical syndromes such as schizophrenia, major depressive disorder (MDD), and bipolar disorder. However, symptoms overlap between some different psychiatric diagnoses, and comparative investigations of psychiatric disorders have revealed both shared and specific genetic[1–3] and environmental risk factors[4,5], and brain transcriptomic profiles[6]. These and other data support a general predisposition to psychopathology or 'p factor' which captures an individual's likelihood of developing any psychiatric disorder[7]. Thus, some genetic and environmental risks

operate trans-diagnostically across multiple psychiatric syndromes, rather than being cis-diagnostically aligned to a specific syndrome, as would be expected if each disorder was a biologically discrete disease entity.

Immune system abnormalities have been observed in case-control studies of many psychiatric disorders, including schizophrenia[8,9], MDD[10], bipolar disorder[11], autism spectrum disorder (ASD)[12], and attention deficit hyperactivity disorder (ADHD)[13]. Among the most consistently reported findings, across multiple disorders, are increased C-reactive protein (CRP)[14], increased pro-inflammatory cytokines[14,15],

[1]Department of Psychiatry, Herchel Smith Building of Brain & Mind Sciences, Cambridge Biomedical Campus, University of Cambridge, Cambridge CB2 0SZ, UK. [2]Cambridgeshire & Peterborough NHS Foundation Trust, Cambridge, UK. [3]Molecular Immunity Unit, University of Cambridge Department of Medicine, Cambridge, UK. [4]Cellular Genetics, Wellcome Sanger Institute, Cambridge, UK. [5]Wellcome Sanger Institute, Wellcome Genome Campus, Cambridge, UK. [6]Open Targets, Wellcome Genome Campus, Hinxton, UK. [7]Human Technopole, Milan, Italy. [8]VA Connecticut Healthcare System, West Haven, CT, USA. [9]Department of Psychiatry, Yale University School of Medicine, New Haven, CT, USA. [10]Departments of Genetics and Neuroscience, Yale University School of Medicine, New Haven, CT, USA. [11]VA San Diego Healthcare System, San Diego, CA, USA. [12]Department of Psychiatry, University of California San Diego, La Jolla, CA, USA. ✉ e-mail: mel41@cam.ac.uk

increased white blood cell counts in both myeloid and lymphoid lineages[16–22], and inflammasome activation[23–25]. Moreover, environmental exposures that elicit an immune response are risk factors for multiple psychiatric disorders, including in utero or parental infections[26,27], childhood and adult infections[28–31], childhood adversity[32], and acute or chronic stress[33]. On this basis, it is conceivable that the immune system could be implicated in the pathogenesis of psychiatric disorders; but the direct evidence for a causal role of immune mechanisms is limited. Longitudinal studies have shown that immune dysregulation can be detected prior to onset of psychiatric disorder[34]; but this could reflect the coincident effects of risk factors for psychiatric disorder, such as high body mass index (BMI), on both the immune system and the brain. Since germline genetic variants cannot be the consequence of disease, sequence variation associated with a disorder (or disorders) could shed light on the immune processes or cells likely to cause mental health symptoms. There is already some genetic evidence that psychiatric risk is mediated by the immune system. Polygenic risk scores (PRS) for depression, bipolar disorder and schizophrenia are associated with increased lymphocyte counts[35]. Immune and psychiatric disorders are genetically correlated[36,37]. Pathway analysis of genes trans-diagnostically associated with schizophrenia, bipolar disorder and MDD implicated neuronal, histone and immune pathways[38]; although a larger trans-diagnostic analysis did not implicate immune cells or pathways[1].

Most genetic variants associated with psychiatric risk are in non-coding regions of the genome, likely exerting their effects by altering the activity of regulatory elements[39] such as promoters or enhancers; and enhancers can be linearly distant (>10 kilobases) from the genes they regulate[40]. Some regulatory elements control gene expression in multiple tissues, but others are specific to particular tissues, or particular cell states. For example, some enhancers are active in stimulated but not resting immune cells[41–43]. The locations and activity status of putative enhancers and promoters in a given tissue can be identified through characteristic epigenetic modifications, such as histone modifications.

Epigenetic mechanisms have long been thought to be important in psychiatry, especially in mediating gene-environment interactions[44,45]. Epigenetic data from brain tissues have been extensively used to investigate the brain cell types and regions implicated by psychiatric risk variants[46–48], by testing whether risk variants tend to be concentrated, or "enriched", in regions of the genome that are active in a given tissue. However, the enrichment of psychiatric risk variants in immune cell subsets has not been extensively explored. Studies to date have tended to use functional information from whole blood or immune organs, which obscures and dilutes possible effects in the myeloid and lymphoid immune cell subsets comprising these samples. There is some evidence of enrichment of risk variants for bipolar disorder in genes characteristic of neutrophils, T cells and haematopoietic stem cells; and for schizophrenia at genes in T cells and chromatin marks in T and B cells[49]. To our knowledge, no studies have demonstrated enrichment of trans-diagnostic risk, or of cis-risk for MDD or ASD, in any immune cell type[1,50–52], or tested if immune cell enrichment is independent of brain tissue enrichment (rather than simply due to coincidental overlap of active genomic regions in brain and immune system cells).

We hypothesized that some genetic risk variants for psychiatric disorders act via their effects on regulatory elements in specific immune cell subsets, thus potentially modulating the response of these cells to infections and other environmental stimuli. We further hypothesized that some of these immunogenetic mechanisms may represent a common pathogenic pathway to multiple psychiatric disorders. To test these hypotheses, we integrated data on common genetic variants associated with trans- and cis-diagnostic risks for psychiatric disorder(s) with data on epigenetically active genomic regions in multiple human cell and tissue types. More formally, we tested the null

hypothesis that a given set of risk variants was not co-located with tissue-specific marks of epigenetic activation more frequently than expected by chance in each of multiple tissues (Roadmap/ENCODE[53,54]), in 19 sorted immune cell subsets (BLUEPRINT[55]), and in ex vivo stimulated naïve and memory CD4+ T cells and macrophages (Soskic dataset[42]). To contextualise our results, we conducted parallel analyses of three "positive control" disorders: Alzheimer's disease, a brain disorder for which genetic risk has been associated with myeloid immune cells[56]; rheumatoid arthritis, a canonical adaptive autoimmune disorder; and body mass index (BMI), a common comorbidity which may contribute to observed immune abnormalities in psychiatric disorders[57]. To our knowledge, this is the first in-depth investigation of the immunological implications of GWAS variants conferring risk for psychiatric disorders.

In this work, we show that trans-diagnostic genetic risk variants for psychiatric disorders, as well as cis-diagnostic risk variants for schizophrenia and depression, are enriched at epigenetically active sites in brain tissues and in lymphoid cells, especially stimulated helper T cells. In contrast, we do not find enrichment of these risk variants in myeloid cells.

## Results

### Trans-diagnostic psychiatric risk is enriched at active chromatin states in T cells

For trans-diagnostic risk of having any one of 8 major psychiatric disorders, we tested for enrichment of genetic risk at active regulatory elements in 88 cells or tissues from the Roadmap consortium, using stratified linkage disequilibrium score regression (s-LDSC). S-LDSC is used to test whether SNP-heritability for a disorder is concentrated or enriched in a genomic annotation[58]. To generate a single binary annotation of active regulatory elements for each tissue, we combined annotations for active promoters and enhancers based on histone marks (see Methods). We found that three main tissue classes were significantly enriched for trans-diagnostic risk variants at regulatory elements, following correction for multiple comparisons: multiple adult and fetal brain regions; T cells; and pancreatic islets (Fig. 1a, Supplementary Fig. 1, Supplementary Data 1).

In the central nervous system (CNS), trans-risk variants were most strongly enriched at regulatory elements in fetal brain tissue samples. There was also significant enrichment ($FDR < 0.05$) at active regulatory elements in brain structures previously reported as abnormal in neuroimaging studies of psychiatric disorders: dorsolateral prefrontal cortex, angular gyrus, inferior temporal lobe, anterior caudate, cingulate gyrus, hippocampus and substantia nigra.

In the immune system, trans-risk variants were significantly enriched ($FDR < 0.05$) at epigenetically active genomic sites in multiple T cell subsets, including cytotoxic, helper and regulatory T cells in adult blood and T cells in cord blood. Conversely, there was no enrichment ($P > 0.05$) of trans-risk in myeloid cells (monocytes, neutrophils). We here use Benjamini-Hochberg correction for multiple testing, as the epigenomic profiles of different cell types are correlated rather than independent.

Many regulatory elements are common to multiple tissues, so we reasoned that this pattern of CNS and immune system enrichment for trans-risk variants could be driven by coincidental overlap of brain and T cell active elements. In this case, the genetic risk would be theoretically expected to have pathogenic effect primarily by its modulation of epigenetically active sites in the brain, with no clearly independent pathogenic role mediated by T cells. We therefore repeated the s-LDSC analysis but included the active annotations for all 10 significantly enriched brain regions as extra terms in the s-LDSC models for every other cell type. In this brain-conditioned analysis, both helper and cytotoxic T cells remained strongly enriched for trans-diagnostic genetic risk (Fig. 1c), while pancreatic islets did not (Supplementary Fig. 2). For enriched immune tissues in the original analysis (at $FDR <$

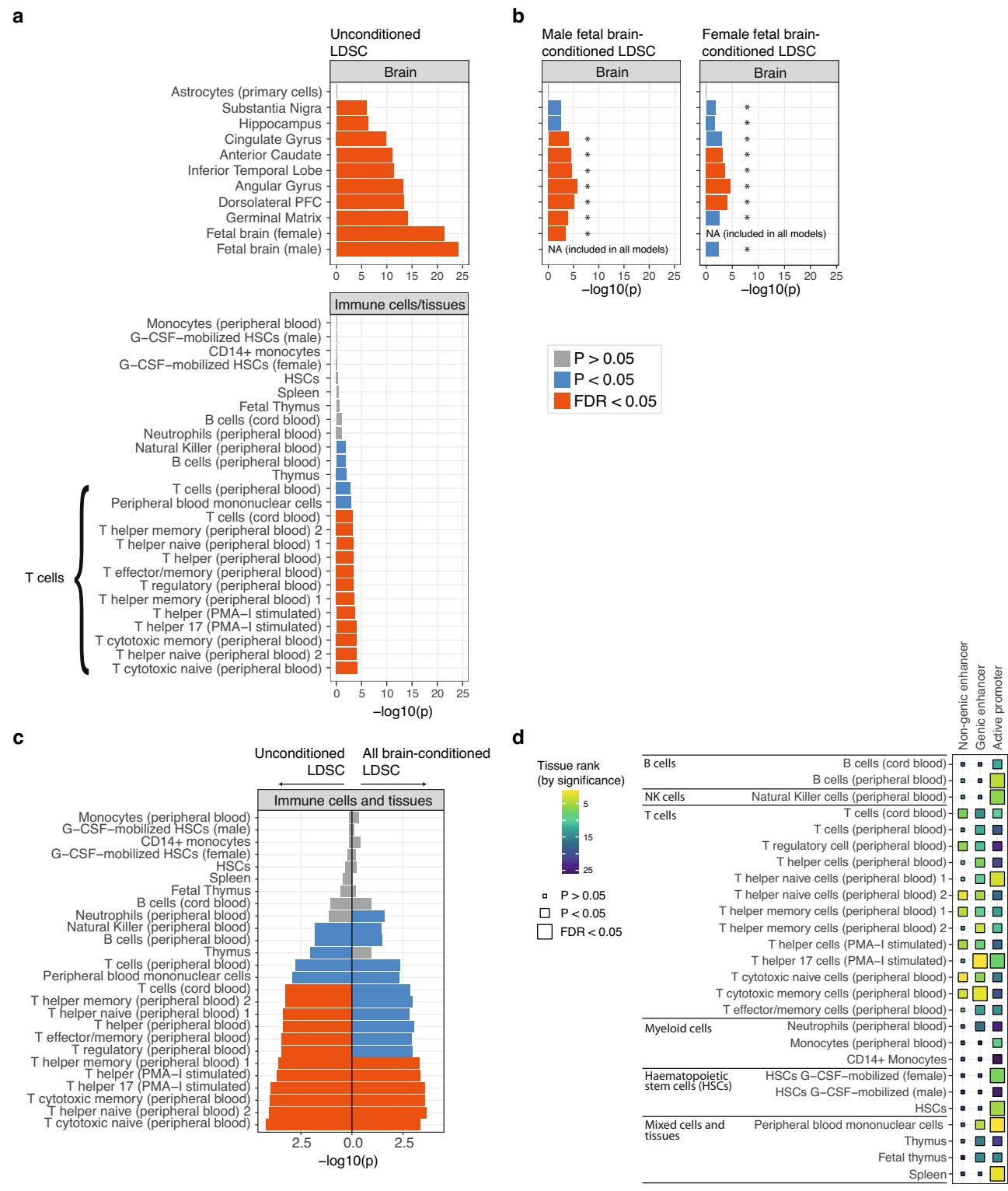

 0.05), none showed significantly decreased enrichment following brain-conditioned analysis (one-sided two-sample *Z*-test, *P* > 0.05). Conversely, including the annotation for male fetal brain (the brain tissue showing strongest enrichment) as an extra term in s-LDSC models significantly reduced trans-risk enrichment in all other brain regions (*Z*-test *P* < 0.05) except the substantia nigra (*P* = 0.09) and hippocampus (*P* = 0.07), reflecting some overlap of active elements between different brain regions at different developmental phases, and validating our statistical approach (Fig. 1b, Supplementary Table 3). We showed the same effect for female fetal brain, the second

most strongly enriched brain tissue (Fig. 1b), excluding a potential effect of sex differences in brain development.

The global active annotation used as a binary marker of epigenetic activation combines trans-risk enrichment at three different classes of regulatory elements: active promoters, genic enhancers (enhancers found in gene bodies), and non-genic enhancers. To identify which classes were most enriched for trans-risk, we tested each class separately and found that the enrichment of trans-risk observed in terms of the global active annotation in T cells was not driven by a single class of regulatory element: there was enrichment of trans-risk at both active

**Fig. 1 | Trans-diagnostic risk enrichment at epigenetically active sites in brain tissue and, independently, in T cells. a** Enrichment of trans-diagnostic risk at active regulatory elements in 88 tissues from the Roadmap epigenomics consortium. *P*-values estimated by stratified linkage disequilibrium score regression (LDSC) analysis (see Methods) were used to test the null hypotheses (one-sided tests) that risk variants were not co-located with epigenetically activated sites more frequently than expected by chance, using the false discovery rate (FDR < 0.05; orange) to correct for multiple tests across *N* = 88 tissues. Tissues with nominally significant enrichment (*P* < 0.05, blue) are also shown for context. For results in all other tissues see Supplementary Fig. 1. **b** Validation of brain-conditioned LDSC modelling. As expected, when the LDSC model for enrichment of adult brain tissues was conditioned on the active regulatory annotations for fetal brain tissue (male and female), there was significant reduction in enrichment across all adult brain tissues (asterisks indicate one-sided two-sample *Z*-tests with *P* < 0.05). **c** Brain-conditioned analysis of enrichment of trans-diagnostic risk variants at active regulatory annotations in immune tissues. Probability of enrichment (log *P* scale) was

estimated by both unconditioned LDSC modelling (left panel of bar chart; same data as in Fig. 1a but on a different *x*-axis range of log probabilities); and brain-conditioned LDSC modelling (right panel of bar chart), one-sided tests. Conditioning enrichment of immune cells on active regulatory annotations in all brain tissues did not significantly reduce enrichment for any immune tissue (all two-sample *Z*-tests had *P* > 0.05); but some T cell subsets were no longer significantly enriched at FDR = 5%; see Supplementary Fig. 2 for comparable results in all other tissues. **d** Enrichment of trans-diagnostic risk in enhancers, genic enhancers and active promoters in all immune subsets (LDSC, one-sided tests). Large tiles show results significant at FDR < 0.05, to correct for the 78 annotations tested; mid-sized tiles show results significant at *P* < 0.05. Tile fill indicates the *P*-value rank within each annotation across cell types. There was enrichment of trans-risk at both enhancers and promoters in multiple adaptive immune cell subsets. See Supplementary Data 1 for full statistics. PFC prefrontal cortex, HSC hematopoietic stem cell, PMA-I phorbol-myristate-acetate and ionomycin.

---

promoters (*FDR* < 0.05) and enhancers (*FDR* < 0.05 for genic enhancers) (Fig. 1c).

## Cis-diagnostic risk is enriched at active chromatin states in T cells

Using data from the Roadmap Epigenomics Consortium, we next investigated the enrichment of cis-diagnostic risk variants at epigenetically active sites in brain tissues and immune cells for each of 5 mental health or neurodevelopmental disorders (schizophrenia, bipolar disorder, MDD, autism and ADHD) and each of 3 positive control disorders (Alzheimer's disease, obesity [BMI], and rheumatoid arthritis).

In the CNS, cis-risks for adult-onset mental health disorders (schizophrenia, bipolar disorder, MDD) were enriched in multiple fetal and adult brain tissues, and cis-risks for child mental health or neurodevelopmental disorders (autism, ADHD) were enriched more selectively in fetal brain tissue. Cis-risk for obesity (BMI) was also enriched for active sites across multiple fetal and adult brain tissues; but cis-risk for Alzheimer's disease was only (nominally) significantly enriched in hippocampus; and cis-risk for rheumatoid arthritis was not enriched in any brain tissue (Fig. 2a, Supplementary Data 1).

In the immune system, similarly to trans-risk, cis-risks for schizophrenia, bipolar disorder, MDD and autism were enriched at globally activated sites in one or more T cell subsets (but with mainly nominal significance *P* < 0.05; Fig. 2b, Supplementary Fig. 3, Supplementary Data 1), with signal driven by both enhancers and promoters (Supplementary Fig. 4a). Cis-risk for rheumatoid arthritis was strongly enriched at globally active sites in multiple immune cell subsets; cis-risk for Alzheimer's disease was significantly enriched in myeloid cells and B cells[56,59]; and cis-risk for BMI was only enriched in one T cell class at *P* < 0.05 (Fig. 2b).

The statistical significance of enrichment results depends partly on the sample size of the underlying GWAS and the heritability and polygenicity of the disorder (factors influencing power, and captured by the SNP-based heritability *Z*-score)[58]; but also on the strength of functional enrichment of the phenotype in that annotation. We hypothesized that, for immune enrichment in psychiatric disorders, the relationship between GWAS power and enrichment might not hold because (a) psychiatric disorders could differ in the degree to which genetic immune factors contribute and (b) immune-relevant genetic risk factors might only be important in a subgroup of patients, and the proportion of the subgroup of total cases would thus affect the immune enrichment detected. Therefore, for the two most enriched immune and brain annotations (naïve cytotoxic and helper T cells; fetal male and female brain), we tested the correlation between heritability *Z*-score and functional enrichment *Z*-score across the 9 disorders included in this study. Strikingly, we found a strong relationship between disorder heritability *Z*-score and detected brain enrichment

(fetal male brain: Spearman's correlation S(7) = 16, *P* = 0.005, *ρ* = 0.87, 95% CI 0.35–1; fetal female brain: S(7) = 16, *P* = 0.005, *ρ* = 0.87, 95% CI 0.35–1), but no correlation between heritability *Z*-score and immune enrichment (cytotoxic T cells: Spearman's correlation S(7) = 120, *P* = 1, *ρ* = 0, 95% CI -0.7–0.7, helper T cells: S(7) = 116, *P* = 0.9, *ρ* = 0.03, 95% CI -0.9–0.8) (see Supplementary Fig. 4b). This suggests that differences in GWAS power are not the primary driver of the different strengths of immune enrichment we observed for different disorders. Differences between disorders in the extent to which immunopathology contributes to symptoms, or the size of the patient subgroup with an immune pathogenesis, may be more important.

## Trans- and cis-risk variants are enriched at active enhancers/promoters in lymphoid cells: BLUEPRINT data

To assess the generalizability of these results in an independent dataset, we tested for enrichment of trans- and cis-risk variants at active enhancer/promoter marks (H3K27ac) in sorted immune cell subsets from the BLUEPRINT consortium[55], using the CHEERS algorithm[42]. The CHEERS algorithm assesses enrichment of genetic risk variants at cell subset-specific epigenetic marks by calculating peak specificity scores, which indicate how specific an epigenetic peak is to that cell type relative to other cell types (see Methods). Cell-type enrichment is calculated as the specificity-weighted sum of overlaps of disease risk variants with these peaks, allowing effects in epigenetically similar cell types to be distinguished. These peak specificity scores are necessarily less correlated across cell subsets than the underlying epigenetic marks, so we here use Bonferroni correction to correct for multiple comparisons, as previously[42]. We replicated our prior key finding from the Roadmap data, i.e., trans-risk was significantly enriched at epigenetically active sites in lymphoid cells; but not myeloid cells (Fig. 3a). We also showed that cis-risk for schizophrenia and depression was significantly enriched after controlling for multiple comparisons (*P*<sub>Bonf</sub> < 0.05) in lymphoid but not myeloid cells (confirming in this dataset the convergent, nominally significant results for these disorders in the Roadmap dataset). The lack of myeloid enrichment was not due to problems with the myeloid data, as we detected the expected enrichment of Alzheimer's Disease risk variants in macrophages (Fig. 3a). As well as T cell enrichment, we also find enrichment of trans-risk and cis-risk for schizophrenia and (especially) depression in B cells, as well as enrichment of trans-risk and cis-risk for schizophrenia in NK cells. For ADHD and bipolar disorder (less well-powered GWAS studies with fewer independent significant loci available for analysis, see Supplementary Table 1), no cell types were enriched at $P_{Bonf}$ < 0.05 (Supplementary Fig. 5a). Despite both schizophrenia and depression showing strong lymphoid enrichment, the specific histone peaks overlapped by risk variants for these disorders were not generally shared between them (Fig. 3b, Supplementary Fig. 4b). This indicates that cis-risks for these two disorders were

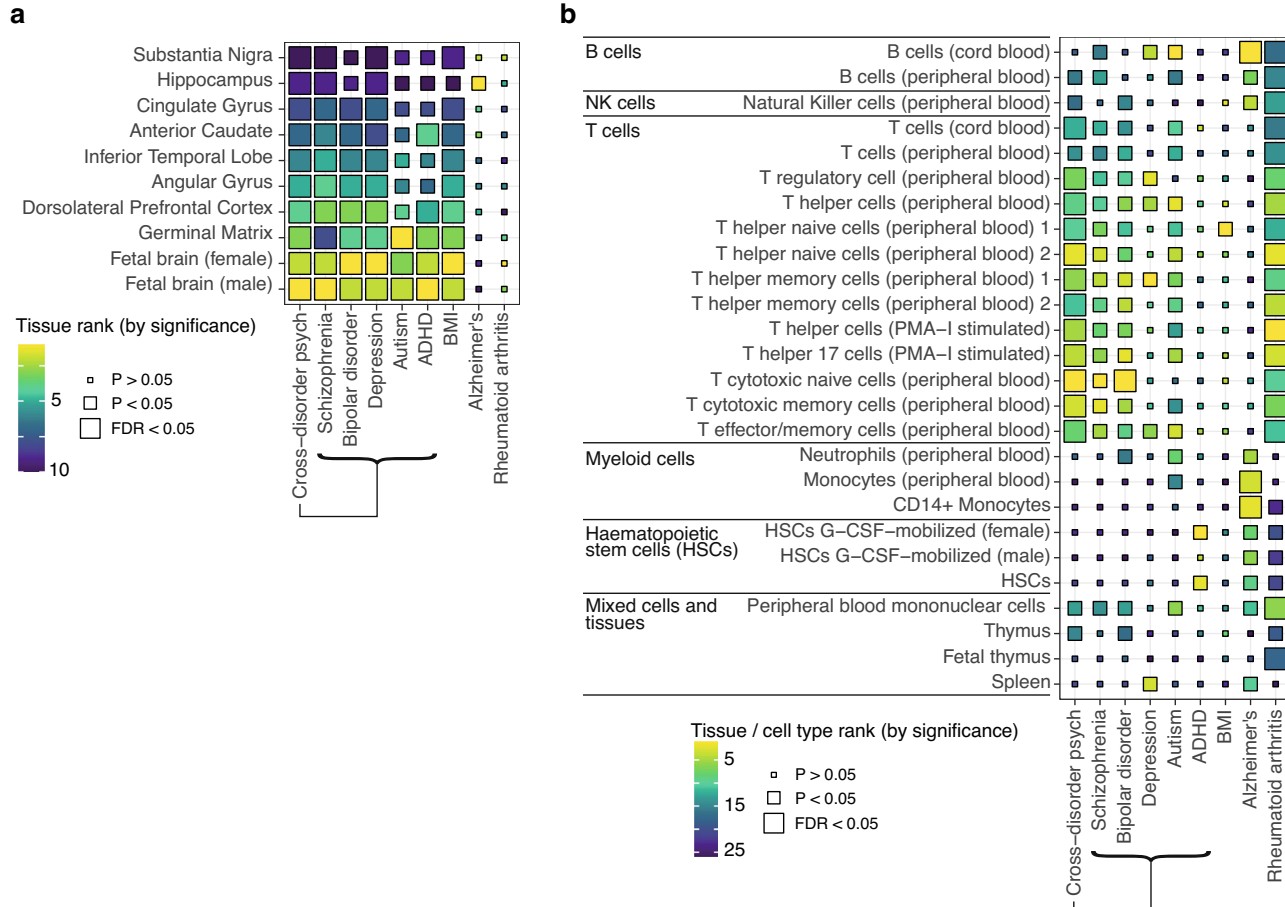

**Fig. 2 | Cis-diagnostic risk enrichment at epigenetically activated sites in adult and fetal brain tissue and immune cells for 8 specific disorders.** For each of 5 mental health disorders (schizophrenia, bipolar disorder, major depressive disorder [MDD], autism, and attention deficit-hyperactivity disorder [ADHD]), and for each of 3 positive control disorders (obesity, Alzheimer's disease and rheumatoid arthritis), enrichment of cis-risk variants at active regulatory elements (active promoters and enhancers) was tested in **a** 10 brain tissue samples (3 fetal) and **b** 26 immune cell subsets and tissues (3 fetal)[54]. *P*-values are shown for the results of stratified linkage disequilibrium score regression (s-LDSC) analysis (one-sided tests), taking the union of active elements in a given cell type as the annotation of interest. Tile size, from large to small, indicates *P*-value thresholds from *FDR* < 0.05 (significant after Benjamini-Hochberg correction for all 88 tissues tested, including those not shown here), through *P* < 0.05 (nominally significant), to *P* ≥ 0.05 (not significant). Tile fill indicates the *P*-value rank within each disorder across all cells/ tissues to facilitate comparisons across results from differently-powered genetic association studies. See Supplementary Fig. 3 and Supplementary Data 1 for full statistics. HSC hematopoietic stem cell, PMA-I phorbol-myristate-acetate and ionomycin, ADHD attention deficit hyperactivity disorder, BMI body mass index.

**Trans- and cis-diagnostic risk variants are enriched at histone-acetylated sites in stimulated T cells: Soskic immune stimulation dataset**

Given that risk of mental health disorders is affected by both genetic variation and environmental factors, we reasoned that trans- and cis-risk variants could be most significantly enriched at sites that were epigenetically activated in immune cells stimulated by cytokines (mimicking environmental insults) towards different activated cell fates. To investigate this hypothesis, and to assess the robustness of our principal findings in a third independent dataset, we used CHEERS to test whether trans- and cis-risks were enriched at cell subset-specific regulatory elements (H3K27ac marks) active during immune cell activation, using a dataset of human naïve and memory CD4+ T (helper) cells and macrophages stimulated ex vivo in the presence of 13 cytokine combinations. The chromatin activity was assessed at early and late timepoints after exposure to cytokine stimulations (16 h and 5 days for T cells and 6 h and 24 h for macrophages), as well as in unstimulated cells[42]. Both trans-diagnostic risk variants, and cis-risk variants for MDD were most significantly enriched in memory T helper cells at day 5 following T cell stimulation with anti-CD3/anti-CD28 beads that mimic activation occurring with T cell receptor-cross-linking; trans-risk variants and cis-risk variants for schizophrenia were also significantly enriched in memory T helper cells at 16 h and in naïve T helper cells at day 5 only (Fig. 4a). The histone acetylation peaks that overlapped with cis-risk variants for MDD in late-activated memory T cells were almost completely disjoint with the peaks that overlapped with cis-risk variants for schizophrenia in late-activated memory T cells (Supplementary Fig. 7a), again demonstrating convergence of enrichment at the immune cell subset level, but divergence at the molecular level of specific regulatory elements. Similarly, although trans-risk and cis-risk for schizophrenia showed the most similar pattern of immune cell enrichment, most of the variant-peak overlaps driving these results were not shared (Supplementary Fig. 7a), implying that trans-risk immune enrichment is not purely being driven by schizophrenia cis-risk variants. Trans-risk enrichment was generally greater for stimulated than unstimulated T cells, with smaller differences in the magnitude of enrichment between different cytokine stimulation conditions (Fig. 4a). For 9 of the 10 cytokine conditions (all except

convergently enriched at a cellular level but distinct at the level of specific regulatory elements. It was also notable that cis-risk variants for obesity overlapped with a set of H3K27ac sites that was largely disjoint with the sets of regulatory elements overlapping with cis-risk variants for psychiatric disorders (Fig. 3b).

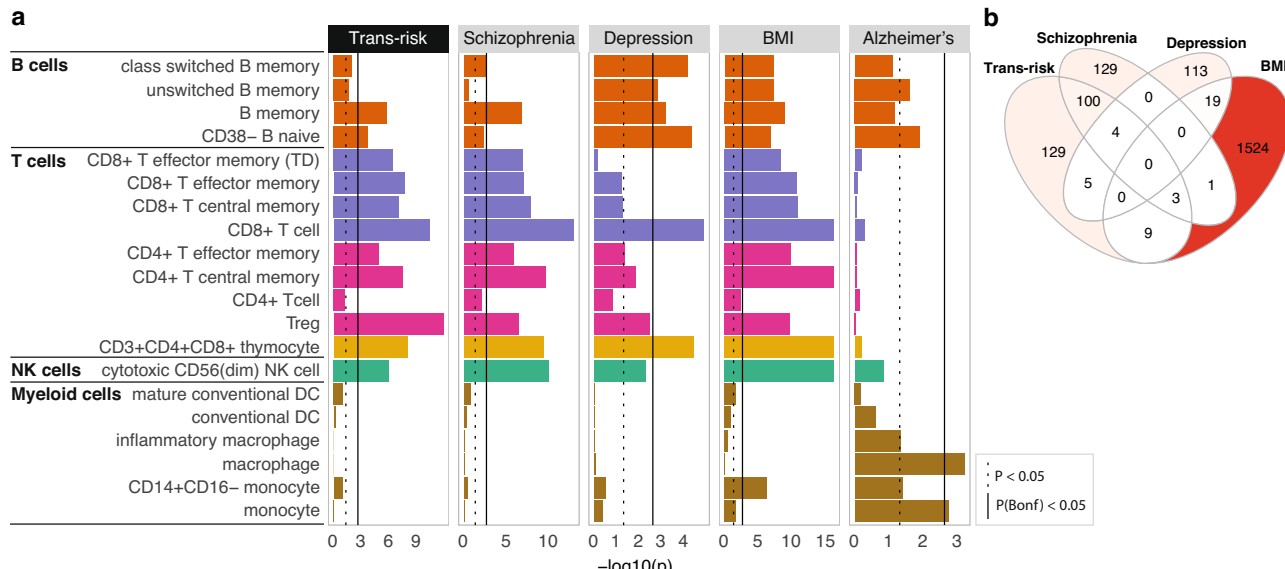

**Fig. 3 | Trans- and cis-diagnostic risk variant enrichment at histone-acetylated marks on adult immune cells in the BLUEPRINT dataset. a** Bar plots show enrichment of genetic risk for each disorder at active promoters/enhancers (H3K27ac marks) in unstimulated, sorted immune cells. CHEERS was used to detect enrichment of risk loci at cell-type specific H3K27ac peaks (see Methods). *P*-values are reported from a discrete uniform distribution (one-sided tests). The dotted black line marks the nominal significance threshold, *P* < 0.05; the solid black line marks the Bonferroni-corrected significance threshold, $P_{Bonf}$ < 0.05. Note differing *x*-axis scales. See Supplementary Fig. 5A for ADHD and bipolar disorder results (non-significant after Bonferroni correction). **b** Venn diagram shows counts of variant-peak overlaps shared between disorders and unique to each disorder (each peak is only counted once even if overlapping multiple variants). For an upset plot of peak overlaps across all disorders, see Supplementary Fig. 5B. TD terminally differentiated, NK natural killer, BMI body mass index.

Th17-cytokine polarizing condition), trans-risk enrichment was significantly greater (*Z*-test, *P* < 0.05) in stimulated compared to unstimulated late-activated memory T cells.

As in the two prior independent datasets, there was no enrichment for trans- or cis-risk of psychiatric disorders at epigenetically activated sites in myeloid cells, either stimulated or unstimulated, with the exception of enrichment of bipolar disorder risk in IL26-stimulated macrophages (Supplementary Fig. 6). Cis-risk variants for obesity were enriched in unstimulated and stimulated T cell states (Fig. 4), but only 9 of the 108 depression-associated H3K27ac peaks also overlapped with BMI risk variants (Fig. 4c, Supplementary Fig. 7b), indicating that cis-risks for obesity and depression were enriched at distinct regulatory elements in the same cell subsets.

For disorders showing enrichment in T cells, we performed pathway analysis (overrepresentation analysis) for those genes overlapping or with transcription start sites nearest to the T-cell specific histone acetylation peaks overlapped by risk variants (although we note that distance-based measures are limited in their ability to link epigenetic peaks with the genes to which they are functionally linked). Trans-risk and cis-risk for schizophrenia showed enrichment of pathways including epigenetic regulation, pre-notch processing, and estrogen-dependent gene expression in T cells, in large part driven by histones and histone-related genes (Fig. 4d and Supplementary Fig. 8). Cis-risk for depression showed enrichment in negative regulation of cold-induced thermogenesis and in dendrite development in T cells. In contrast, rheumatoid arthritis showed enrichment of lymphoid cell differentiation, activation, and response to antigenic stimulus (Supplementary Fig. 8). Notably, most of the T cell genes highlighted by the epigenetic analysis of trans-risk, or cis-risks for schizophrenia and depression (see Supplementary Data 2), did not feature in any enriched pathways, perhaps because the immunobiology relevant to psychiatric disorders has not yet been captured in pathway databases.

We note that many of the T-cell specific histone acetylation peaks in the Blueprint data which were co-located with risk variants were not also overlapped by T-cell specific peaks in the Soskic dataset co-located with risk variants (55% for MDD and 54% for schizophrenia).

Likewise, many of the Soskic T-cell peaks co-located with risk variants were not also overlapped by any of the Blueprint T-cell peaks co-located with risk variants (36% for MDD and 29% for schizophrenia). This suggests that the replicable T cell enrichment observed was not driven exclusively by similarities between the specific peaks detected in the different datasets.

## Enrichment of risk for MDD and schizophrenia at active regulatory elements in T cells shows convergence at the cellular scale, but with limited convergence at the molecular scale

For both the Blueprint and Soskic datasets, both MDD and schizophrenia risk variants were enriched in T cells at H3K27ac histone acetylation marks. There were no significant differences between disorders in terms of the classes of regulatory elements involved (the proportions of promoters, genic enhancers and non-genic enhancers at the implicated peaks), or in terms of the genomic distance between each implicated acetylation peak and the nearest transcriptional start site (see Supplementary Table 4). However, this convergence between disorders at a cellular scale was not representative of convergence at the molecular scale of the acetylation peaks overlapped by cis-diagnostic risk variants, which were not generally shared between disorders. For example, there were only two (of 211 total) T cell acetylation peaks implicated in common between MDD and schizophrenia in the Blueprint dataset, and three (of 214 total) in the Soskic dataset. The genes implicated by cis-diagnostic variant-peak overlap in T cells were also largely discordant between disorders. Only 5 genes were consistently implicated in both MDD and schizophrenia: in both the Blueprint and Soskic datasets, *COA8/APOPT1* (a proapoptotic mitochondrial protein) and *MAD1L1* (a checkpoint protein); and in the Soskic dataset only, *FURIN* (a protease), *SNORD18* (a non-coding RNA), and *RP11-73M18.2* (a kinesin light chain). We also compared the cis-diagnostic GWAS statistics independently estimated for MDD and schizophrenia at each variant that was co-located with MDD-schizophrenia discordant acetylation peaks (peaks implicated in one but not both disorders). This analysis demonstrated that discordance of acetylation peaks was not simply reflective of sub-genome wide

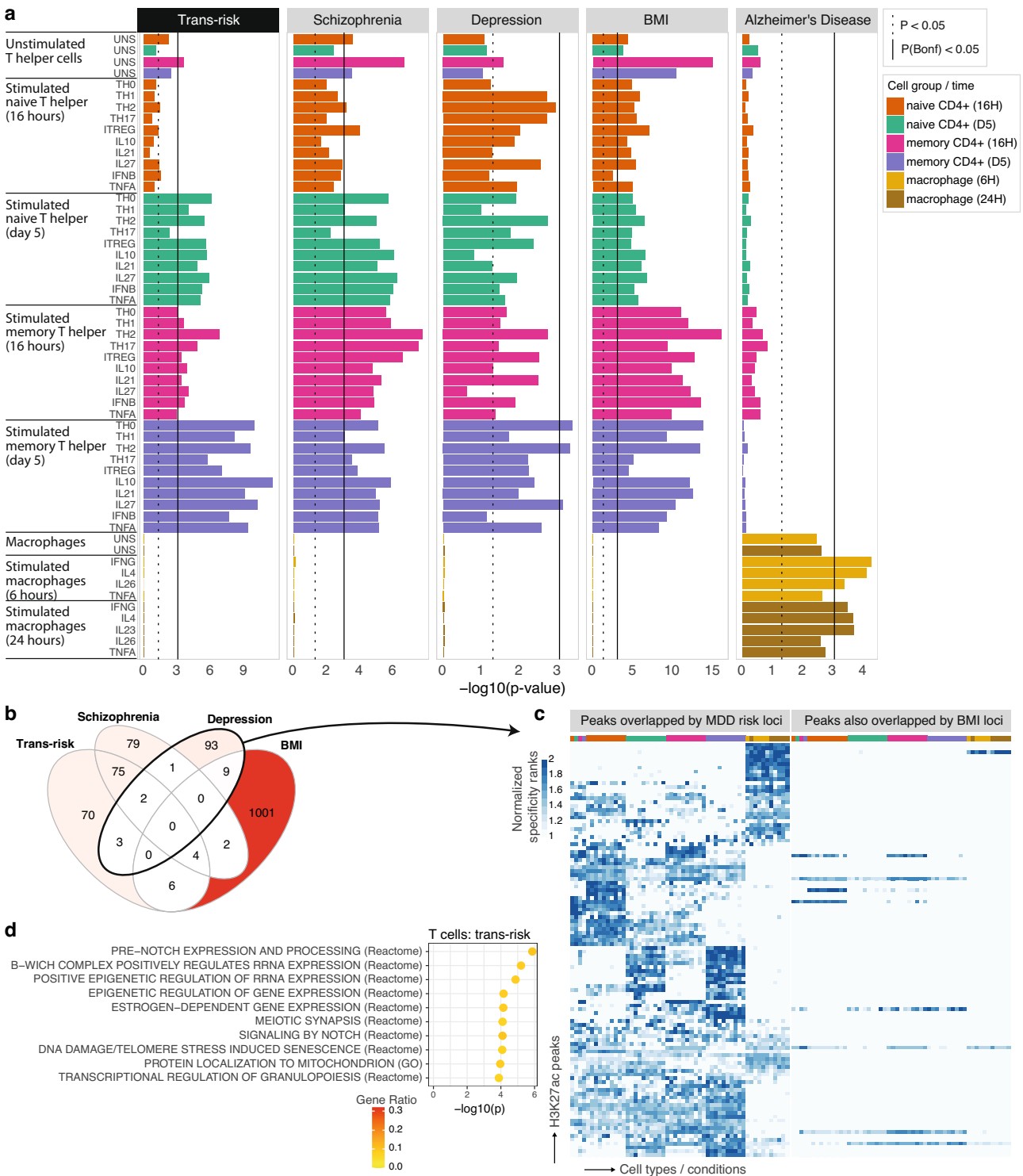

**a** (bar chart panels: Trans-risk, Schizophrenia, Depression, BMI, Alzheimer's Disease; x-axis −log10(p-value))

Legend:
- P < 0.05
- P(Bonf) < 0.05

Cell group / time:
- naive CD4+ (16H)
- naive CD4+ (D5)
- memory CD4+ (16H)
- memory CD4+ (D5)
- macrophage (6H)
- macrophage (24H)

**b** Venn diagram (Trans-risk, Schizophrenia, Depression, BMI)

**c** Peaks overlapped by MDD risk loci / Peaks also overlapped by BMI loci; Normalized specificity ranks; H3K27ac peaks; Cell types / conditions

**d** T cells: trans-risk
- PRE–NOTCH EXPRESSION AND PROCESSING (Reactome)
- B–WICH COMPLEX POSITIVELY REGULATES RRNA EXPRESSION (Reactome)
- POSITIVE EPIGENETIC REGULATION OF RRNA EXPRESSION (Reactome)
- EPIGENETIC REGULATION OF GENE EXPRESSION (Reactome)
- ESTROGEN–DEPENDENT GENE EXPRESSION (Reactome)
- MEIOTIC SYNAPSIS (Reactome)
- SIGNALING BY NOTCH (Reactome)
- DNA DAMAGE/TELOMERE STRESS INDUCED SENESCENCE (Reactome)
- PROTEIN LOCALIZATION TO MITOCHONDRION (GO)
- TRANSCRIPTIONAL REGULATION OF GRANULOPOIESIS (Reactome)

Gene Ratio: 0.0, 0.1, 0.2, 0.3; x-axis −log10(p)

significance of association signals in the disorder where variant-peak overlap was not detected. In fact, as shown in Supplementary Fig. 9, many of the variants that were co-located with discordant acetylation peaks had different signs (negative vs. positive) or strengths of association with the two disorders.

## Discussion

We examined the enrichment of genetic risk variants for psychiatric disorders at epigenetically activated regulatory sites across multiple tissues. As expected, trans-diagnostic risk variants, commonly associated with multiple mental health and neurodevelopmental disorders, were significantly enriched at active regulatory sites in several adult and fetal brain tissue samples. Strikingly, we also found that trans-diagnostic risk variants were significantly enriched at an independent set of regulatory elements in peripheral blood lymphoid cells (but were not enriched in myeloid cells). Our key results—enrichment of trans-risk in T cells and lack of enrichment in myeloid cells—were statistically robust to multiple comparisons and replicated in three independent datasets, suggesting a previously unknown effect of trans-diagnostic genetic risk on T cells. Other lymphoid cells (for which fewer datasets were available) are likely also implicated in pathogenesis, as we also found enrichment of trans-risk in B cells and NK cells.

**Fig. 4 | Trans- and cis-risk variant enrichment at histone-acetylated marks on experimentally stimulated immune cells in the Soskic immune stimulation dataset. a** Bar plots show enrichment of genetic risk for each condition at active promoters/enhancers (H3K27ac marks) in sorted and unstimulated or ex vivo stimulated immune cell subsets: macrophages, naïve CD4$^+$ (helper) T cells and memory CD4$^+$ T cells, assayed at both early and late timepoints after stimulation with one of several different cytokine cocktails promoting differentiation to different T cell states (as shown in row labels). CHEERS was used to detect enrichment of risk loci at cell-type specific H3K27ac peaks (see Methods). *P*-values are reported from a discrete uniform distribution (one-sided tests). The dotted black line marks the nominal significance threshold, *P* < 0.05; the solid black line marks the Bonferroni-corrected significance threshold, $P_{Bonferroni}$ < 0.05. Note differing *x*-axis scales. Results for other disorders are shown in Supplementary Fig. 6. **b** Venn diagrams show counts of variant-peak overlaps shared between disorders and unique to each disorder. For an upset plot of peak overlaps across all disorders, see Supplementary Fig. 7b. **c** All Soskic immune stimulation dataset peaks overlapped by risk variants for major depressive disorder (MDD). Each row corresponds to an H3K27ac peak overlapping a risk variant for MDD; each column corresponds to a different cytokine-induced cell state, ordered and colored as in Fig. 4a (see legend). The blue fill shade represents how specific each peak is to each cell state (specificity rank of each peak normalized to the mean specificity rank of all peaks). Only 9 of the 108 MDD-associated H3K27ac immune peaks also overlap BMI risk variants. **d** For peaks which were both highly specific to T cells (including both unstimulated and stimulated cells) and overlapped by trans-risk variants, nearest genes were identified and tested for enrichment for curated biological pathways (GO and Reactome) using a one-sided hypergeometric test. Only the 10 most significant pathways are shown (all *FDR* < 0.05). Fill colour indicates gene ratio (number of test genes in the pathway/total number of test genes). See Supplementary Fig. 8 for results for cis-diagnostic risks. GO gene ontology, BMI body mass index.

Further investigation of cis-diagnostic risk variants, specifically associated with one of 5 mental health or neurodevelopmental disorders, confirmed significant enrichment of genetic risks for schizophrenia and major depressive disorder at active promoters and enhancers in peripheral lymphoid cells (but not myeloid cells). Epigenetically activated sites in T cells, especially cytokine-stimulated CD4$^+$ T cells, were most consistently and significantly enriched for cis-diagnostic variants associated with either schizophrenia or MDD. However, at a molecular scale, the active regulatory elements co-located with these cis-diagnostic variants were largely specific to each disorder. Many of the variants driving T cell enrichment in one disorder were not associated with the other disorder, even at a nominal significance threshold. This suggests convergence of risk for schizophrenia and depression at a cellular level in the immune system, i.e., activated T cells, and raises questions about how epigenetic activation at disorder-specific risk variants might relate to the different clinical phenotypes or pathogenic pathways of schizophrenia and depression. We also found strong enrichment of risk for depression in both naïve and memory B cells. To our knowledge, this is the first demonstration of enrichment of genetic risk for MDD at epigenetically active sites in lymphoid cells (or indeed any immune cell type). Notably, in all three datasets, immune enrichment of schizophrenia risk variants was much greater than for depression risk variants, despite the larger size of the depression GWAS dataset.

The cis-diagnostic enrichment results for schizophrenia and MDD were statistically robust to multiple comparisons and in clear contrast to the comparable results for 3 positive control disorders. Cis-risks for Alzheimer's disease were significantly enriched at epigenetically activated sites in myeloid cells (but not lymphoid cells); cis-risks for rheumatoid arthritis were enriched at active sites in myeloid and lymphoid cells (but not brain tissue); and cis-risks for obesity (BMI) were enriched at active sites in brain tissue and (in some analyses) in immune cells, but with effects on regulatory elements distinct from those implicated by psychiatric disorders.

On this basis, we propose that genetic variants associated with increased risk for psychiatric disorders are likely to interact with epigenetic activation of specific and distinct regulatory elements in both the central nervous system and the adaptive immune system. This hypothesis-generating work immediately raises three key questions. What environmental exposures cause epigenetic modification at risk-enriched sites in T cells? How could atypical T cell phenotypes cause changes in the CNS that are ultimately manifest as mental health or neurodevelopmental disorders? What are the antigen presenting cells (our data suggest they may be B cells) which activate atypical CD4$^+$ T cells?

Infection is the most likely environmental stimulus to induce epigenetic activation in the immune system. There is also increasing evidence that psychosocial stress, especially in early life, can cause epigenetic activation of glucocorticoid receptor-related genes in animal models; and early life adversity has been associated with long-term changes in blood immune biomarkers in human longitudinal studies[45]. However, here we focus on the abundant epidemiological evidence that fetal and post-natal infections increase the risk for multiple psychiatric disorders[26,27,30,31]. The immune mechanisms by which early-life infection predisposes to later psychiatric symptoms are not known. But we do know that fetal or childhood infections can cause long-term changes in adaptive immune cell phenotypes, including T cell memory of antigens and B cell production of antibodies, that are crucial to development of adult immunity[60]. Thus, it is conceivable that the epigenetically activated sites enriched for trans- and cis-risks in T cells and memory B cells in these data were "marked" by exposure to infection or inflammation; and that genetic risk variants modulate the infection-induced activation of regulatory elements, leading to atypical T or B cell phenotypes following infection in people at genetic risk of psychiatric disorder. There is already some epidemiological evidence for gene-by-environment interactions between infection and risk variants for schizophrenia[61–63] and MDD[64]. Many aspects of our data are compatible with this concept. For example, our finding that trans- and cis-diagnostic risk variants were enriched at sites epigenetically activated by delayed T cell responses to a wide range of pro-inflammatory cytokine stimuli seems consistent with the epidemiological finding that increased risk of multiple psychiatric disorders is found following a wide range of different infections[12,28,29,65,66].

Atypical T cell phenotypes could conceivably have effects on the brain by at least two broad routes: via stimulus-driven T cell activation and via developmental pathways (Fig. 5). Atypical T cells may impact on neuronal function via soluble inflammatory mediators[67,68]; via contact-dependent mechanisms[69]; or via their effects on other immune or non-immune cells which in turn affect neurons[69]. Developmentally, T cells have an important physiological role in controlling microglial phagocytosis of synaptic terminals and neurites as part of normal childhood and adolescent neurodevelopmental programs of synaptic pruning[70]. Thus atypical T cells in the meninges or brain could lead, via atypical synaptic pruning[70,71], to the disrupted brain connectivity seen in schizophrenia and other psychiatric disorders[72].

In contrast with autoimmune diseases, which tend to show greatest enrichment in early T cell activation states[42], the strongest enrichment for psychiatric risk variants was in T cells, especially late-activated memory CD4$^+$ T cells, and memory B cells. This may reflect abnormalities in the resolution (rather than onset) of immune responses to infection or social stress, potentially leading to chronic, low-grade peripheral inflammation seen in many psychiatric disorders[14,15].

Surprisingly, given the prior focus on innate immune abnormalities associated with psychiatric disorders, epigenetically active sites in myeloid cells were not significantly enriched for trans-risk variants or for cis-risk variants for schizophrenia or MDD. What does this mean for the pathogenic role of myeloid cells in these disorders? It may be that genetic risk variants are indeed enriched at epigenetically active sites

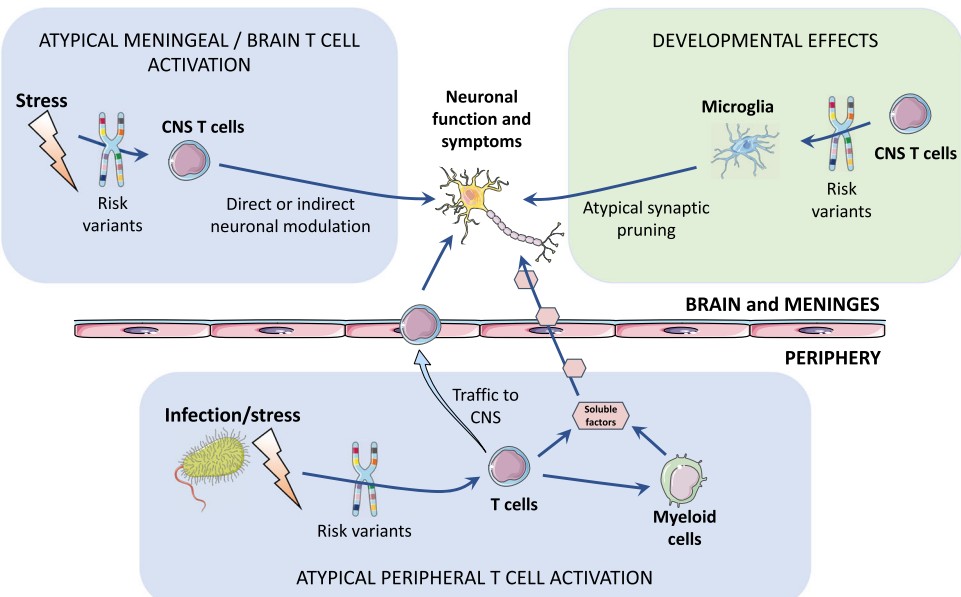

**Fig. 5 | Schematic of potential pathogenic pathways by which genetic risk variants enriched at epigenetically active sites in T cells could lead to neuronal changes and ultimately psychiatric disorders.** Infection or other stressors may induce activation of regulatory elements in T cells that are enriched for trans- or cis-diagnostic risk variants, potentially leading to atypical T cell phenotypes and downstream activation of innate immune (myeloid) cells in the periphery and CNS (light blue boxes). Atypical activation of T cells resident in the CNS, or trafficking into the meninges and brain from the periphery, could adversely affect neuronal function. Developmentally (light green box), T cells are known to control microglial pruning of neuronal synapses as part of normative brain developmental programs in childhood and adolescence. Atypical T cells, in genetically-at-risk individuals, could promote atypical microglial pruning of synapses, contributing to the formation of disconnected networks or circuits in the adult brain. CNS, central nervous system. Parts of the figure were drawn by using pictures from Servier Medical Art. Servier Medical Art by Servier is licensed under a Creative Commons Attribution 3.0 Unported License (https://creativecommons.org/licenses/by/3.0/).

in myeloid cells, but only under stimulation conditions not represented in the three datasets we analysed. Alternatively, it could be that myeloid abnormalities seen in psychiatric disorders are downstream of epigenetically activated risk variants in lymphoid cells, or are driven by entirely environmental factors, rather than by genetic factors or gene-by-environment interactions.

The statistical significance of results for risk variant enrichment at epigenetically active regions reflects in part the sample size of the GWAS datasets used and the heritability and polygenicity of the disorders. However, in contrast to our results for brain enrichment, we did not find any correlation between GWAS statistical power and immune enrichment. This suggests that while GWAS power can explain some differences between disorders in the significance of functional enrichment (as in the brain), differences in immune cell enrichment may in part reflect how frequently immune mechanisms are implicated in individual patients clinically diagnosed with a specific disorder. Thus differences between disorders in the strength of immune enrichment seen here may be more indicative of between-disorder differences in how strongly immune mechanisms contribute to pathogenesis in general, or what proportion of cases have an immune pathogenesis. The immune enrichments we detected were significantly weaker than enrichments in brain tissues—this may reflect a weaker pathogenic contribution of epigenetically activated risk variants in the immune system (compared to the brain); or it may be that genetic immune mechanisms can have a larger effect but only in a subgroup of patients. The genetic architecture of psychiatric disorders is currently incomplete. As more risk variants are identified in future, the number of epigenetically active loci implicated in adaptive immune cells will likely increase, and understanding of their functional implications will be further refined. However, analyses based on alternative European GWAS datasets are unlikely to alter our major findings, which focus mainly on patterns of enrichment across different cellular subsets for risk variants that have already been significantly associated with one or more psychiatric disorders. In short, we expect our current results to

represent a robust core set of acetylated regions in T cells which will be enhanced rather than undermined by future increase in the scale and dimensionality of GWAS studies in psychiatry.

We focused here on European ancestry genetic results, as the currently available datasets are from European participants, but the immunogenetics of psychiatric risk should be examined in other ancestries. In addition, the epigenetic datasets used here are predominantly adult: given the role of developmental insults in psychiatric risk, it will be important to investigate genetic enrichment in immune cells sampled at different developmental stages, including adolescence. Likewise the immune cell states considered in this analysis are the canonical states associated with infection and auto-immunity. It will also be important to explore whether genetic risk variants modulate immune cell phenotypes induced by exposure to non-infectious environmental stimuli e.g. stress, especially given that childhood adversity and other social stressors are known to profoundly increase risk for multiple psychiatric disorders[73]. We also know that many psychiatric conditions show sex differences in prevalence, and we presented enrichment results separately for male and female fetal brains; but most tissue classes included in the multi-tissue epigenetic datasets we analysed were not represented by sex-stratified data. In particular, appropriate sex-stratified epigenetic data on immune cells were not openly available, although such data would likely be informative in further analysis of risks for neuropsychiatric disorders.

In psychiatry, T cell phenotypes have been most investigated in schizophrenia[74], with some evidence of decreased proliferative responses to stimulation[75]. Our findings motivate further investigation of T cell and B cell phenotypes across multiple psychiatric conditions, with a focus on how trans-risk affects activated T cells (e.g., stimulated ex-vivo). Functional genomic analysis of T cell subsets from patient cohorts will be particularly important to directly test for disease-associated alterations in DNA accessibility, histone modifications, enhancer-promoter interactions and gene expression. We hypothesize

that these alterations will be found at those T cell peaks identified in our analysis as overlapping risk variants. However, there may be broader epigenetic consequences of risk variants, especially given that pathway analysis implicated epigenetic regulation processes, which may occur at sites distant from the risk variants.

In conclusion, genetic risk variants for psychiatric disorders were significantly enriched at epigenetically active enhancers/promoters in adaptive immune cells, especially stimulated T cells. This suggests a mechanistic role for T cells in the pathogenesis of multiple psychiatric disorders, hypothetically by mediating the interaction between environmental exposures to biological or social threats and genetic risk variants.

## Methods

### Trans- and cis-diagnostic genetic risk variants for psychiatric disorders

The primary GWAS datasets used for the identification of trans- and cis-risk genes are listed in Supplementary Table 1. We used summary statistics from a meta-analysis of trans-diagnostic risk across 8 mental health or neurodevelopmental disorders[1]: ASD, bipolar disorder, MDD, obsessive-compulsive disorder, schizophrenia, anorexia nervosa, ADHD, and Tourette syndrome. For analysis of cis-risk, i.e. risk of a specific psychiatric disorder, we separately tested 5 large primary genome-wide association studies (GWAS) of MDD[76], bipolar disorder[52], schizophrenia[77], autism[78], and ADHD[79]. For comparative purposes, we analysed GWAS results for BMI[80], Alzheimer's disease[81], and rheumatoid arthritis[82]. For all disorders except MDD, we selected the largest publicly available, predominantly-European GWAS dataset; for MDD, we used a larger recent European GWAS[76].

### Testing for enrichment of genome-wide genetic risk at regulatory elements (ROADMAP data)

Stratified linkage disequilibrium score regression (abbreviated as LDSC throughout this paper) can be used to test whether genetic risk is concentrated or enriched in a genomic annotation, e.g., a set of active regulatory elements in a specific cell type[58]. S-LDSC hinges on the fact that the disease association statistic for a given genetic variant depends on whether that variant is linked to the disease, but also whether variants in linkage disequilibrium (LD) with that variant are linked to the disease. By testing whether variants in LD with the annotation of interest tend to have higher association scores than variants elsewhere, we can calculate an enrichment score capturing the tendency of SNP-based heritability for that disease to be co-located with that annotation[58]. We used this method to test for enrichment of psychiatric risk variants at active regulatory elements in 88 cell or tissue types.

For a given tissue, CHiP-seq data assaying multiple histone marks can be integrated to segment the genome into annotations representing different functional epigenetic states, e.g., enhancers, promoters, repressed regions[54]. The IDEAS algorithm[83] leverages shared features across cell types to improve this segmentation. Lacking a strong prior hypothesis about which particular regulatory elements in immune cells would be implicated by psychiatric risk, we generated a simple functional annotation of active states for each tissue in the Roadmap Epigenomics Dataset, which includes samples from all major organ systems including brain, heart, muscle, gut, adipose, skin, reproductive and immune tissues[54]. Data for a given tissue or cell type sometimes come from multiple donors—as is the case for most of the brain and immune samples—and sometimes from single donors (see https://egg2.wustl.edu/roadmap/web_portal/meta.html for metadata). Immune cell subsets were magnetically sorted from live donor blood samples; brain tissues were homogenized post-mortem samples. For each Roadmap tissue/cell type, we generated a whole genome binary annotation of active regulatory elements (Fig. 1a, Supplementary Fig. 1) from IDEAS annotations based on 5 epigenetic histone marks

(H3K4me3, H3K4me1, H3K36me3, H3K27me3 and H3K9me3). We combined the 6 IDEAS annotations representing active promoters and enhancers to generate a single binary annotation of active regulatory elements for each tissue. More exactly, we merged the IDEAS annotations for active transcription start sites (10_TssA); regions flanking active TSS (8_TssAFlnk); weak TSS (14_TssWk); enhancers (4_Enh); genic enhancers (6_EnhG); and genic enhancers associated with transcription (17_EnhGA), following a previous definition of active states[84]. We generated partitioned linkage disequilibrium (LD) scores for each tissue as recommended, using HapMap3 SNPs[58].

We then used s-LDSC to test the enrichment of psychiatric risk variants in each cell type, using a separate model for each cell type, as is standard. Summary statistics were preprocessed using the LDSC recommended script munge_sumstats.py and we performed s-LDSC for each tissue in the Roadmap dataset, using recommended settings, excluding the extended MHC region (GRCh37 chr6:25-34 Mb). The P-values for s-LDSC are one-sided tests that the regression coefficient corresponding to the cell type specific annotation of interest is greater than zero. The regression coefficient corresponds to the change in per-SNP heritability due to a given annotation beyond that explained by the baseline model and other annotations and can be interpreted as the effect size for that annotation. P-values were corrected for multiple comparisons across tissues using Benjamini-Hochberg false discovery rate. We coloured heatmaps by P-value rank to aid comparison across disorders or annotations which are differently powered.

We ran the analyses for male and female fetal brain separately as these sex-stratified datasets were provided separately by the Roadmap Epigenomics Consortium (Roadmap Epigenomics Consortium et al. 2015 Nature) and our analysis of sex-stratified data was compatible with prior multi-tissue analyses of Roadmap data[49,83]. All analyses were thus conducted and reported using tissue classes, sex-stratified for some but not all tissues, as defined by Roadmap (the full list is shown in Supplementary Fig. 1).

To further dissect the s-LDSC results for the active annotations, we also performed s-LDSC for the 3 types of genomic element comprising the active annotation: promoters, enhancers, and genic enhancers. We generated partitioned LD scores for the promoters (10_TssA, 8_TssAFlnk and 14_TssWk), enhancers (4_Enh) and genic enhancers (6_EnhG and 17_EnhGA) (Fig. 1d, Supplementary Fig. 4a) then performed s-LDSC using default settings for each of these annotations in the Roadmap immune tissues.

To account for the possible confounding effect of shared regulatory elements between brain and immune tissues, we also performed brain-conditioned enrichment analyses: for each tissue's s-LDSC model, we added terms for the active regulatory annotations for possibly confounding brain regions. In the LDSC model, the coefficient $\tau$ (which captures the contribution to SNP-heritability) for a given genomic category/annotation ($C$) is estimated by regressing $\chi^2$ (the SNP association statistics) against $\ell(j,C)$ (the linkage disequilibrium score for SNP $j$ with respect to category/annotation $C$):

$$E[\chi^2] - \sum_{C=1}^{n} \tau_C \ell(j,C) \qquad (1)$$

For the original s-LDSC models, the annotations ($C$) included in each multiple regression were the cell specific annotation of interest plus the standard non-cell type specific annotations (baseline v1.2, see https://storage.googleapis.com/broad-alkesgroup-public/LDSCORE/readme_baseline_versions). For the brain-conditioned models, the categories in each regression additionally included the annotations for the potentially-confounding brain regions: e.g., in Fig. 1c, the 10 significantly enriched brain annotations from Fig. 1a were also included in the s-LDSC model for each immune annotation.

SNP heritability Z-scores (heritability / standard error) and s-LDSC Z-scores (enrichment coefficient / standard error) were estimated

using LDSC. Confidence intervals for Spearman correlations were generated by bootstrapping (10,000 replicates). To compare the results of the original and brain-conditioned analyses, we used a one-sided two-sample $Z$-test as follows, where $\beta_1$ is the coefficient for the annotation in the original analysis and $\beta_2$ is the coefficient in the brain-conditionedal analysis. SE is the standard error of the coefficient for the original ($SE_1$) or conditional ($SE_2$) analysis. $Z$-scores were converted to $P$-values.

$$Z = \frac{\beta_1 - \beta_2}{\sqrt{(SE_1)^2 + (SE_2)^2}} \qquad (2)$$

### Testing for enrichment of genetic risk variants in cell-type specific active promoters/enhancers

To compare enrichment of genetic risk at regulatory marks in different immune cell subsets, and immune cells stimulated under different conditions, we used the CHEERS algorithm[42]. CHEERS quantifies the overlap of lead (independently significant) genetic risk variants with cell-specific epigenetic peaks. Crucially, CHEERS facilitates the comparison of similar cell types or conditions, which tend to have similar epigenetic profiles, by calculating peak specificity scores, indicating how specific a peak is to that cell type relative to other cell types, then quantifying cell-type enrichment as the specificity-weighted sum of overlaps of disease risk variants with these peaks. While s-LDSC leverages genome wide-information, CHEERS focuses on risk loci which meet genome-wide significance. In brief, CHEERS identifies histone acetylation peaks (or other genomic annotations) which overlap lead variants or variants in strong LD ($r^2 > 0.8$) with lead variants; then calculates the mean cell type specificity score (in that cell type) of those peaks, which captures the degree of enrichment of that cell type for a given disorder. Seeking overlap between regulatory elements and any variant in the LD block of a given lead variant ensures that the CHEERS method is not sensitive to subtle differences in tag variants between different association studies. One-sided $P$-values were reported from a discrete uniform distribution (reflecting the ranking of specificity scores within each cell type) and corrected for multiple comparisons across tissues using a Bonferroni correction. To identify lead disease risk loci, all summary statistics were processed consistently: liftover to hg38, harmonization, removal of MHC region, and distance-based clumping (see below for more detail). We applied CHEERS using two human H3K27ac ChIP-seq datasets: (i) BLUEPRINT consortium data from 19 sorted unstimulated immune cells subsets (see Fig. 3a)[55] and (ii) the Soskic immune stimulation data from sorted and ex vivo stimulated immune cells[42]. H3K27ac marks active (rather than inactive or poised) enhancer and promoter regions[53,85]. In the Soskic immune stimulation experiment, macrophages, naïve CD4+ T cells and memory CD4+ T cells were stimulated using different cytokine cocktails associated with autoimmunity or known to promote different cell fates (see Fig. 4a). In addition, generic T cell receptor and CD28 co-stimulation signals were provided in all stimulated T cell conditions using beads coated with anti-CD3 and anti-CD28 antibodies. H3K27ac data were processed[42] to obtain cell-type specificity scores for H3K27ac peaks in each cell type or state. Here, we ran CHEERS using $r^2$ linkage disequilibrium values taken from unrelated European individuals from the 1000 genomes dataset[86], calculated using PLINK[87].

To compare the enrichment between stimulated and unstimulated cell subsets, we used a one-sided, two-sample $Z$-test as follows, where $x_1$ is the mean specificity rank for the stimulated cell subset and $x_2$ is the mean specificity rank for the corresponding unstimulated cell subset. SE is the standard error of the mean and depends on the number of variants overlapping peaks. For a given disorder, SE is the

same across different annotations, as peaks are called across the dataset as a whole. $Z$-scores were converted to $P$-values.

$$Z = \frac{x_1 - x_2}{\sqrt{2(SE)^2}} \qquad (3)$$

The three epigenetic datasets we used have some overlap in the immune cell subsets represented. For example, CD4+ T cells were represented in all three datasets; monocytes and B cells were represented in both the Roadmap and Blueprint datasets; and macrophages were represented in both the Roadmap and Soskic datasets. However, the annotations used for the Roadmap analysis ("active states", comprising enhancers and promoters defined using multiple histone marks) are different from the annotations used for the Blueprint and Soskic analyses (cell subtype-specific H3K27ac marks).

### Identification of independent risk loci

To identify independently significant loci for each disorder, we reprocessed all summary statistics consistently. Given the lack of well-matched linkage disequilibrium data for the populations underlying these studies, we aimed to conservatively identify independent lead variants without using LD information or conditional analysis within loci. We first lifted over the summary statistics (autosomal chromosomes only) and harmonized variants to the reference strand using the EBI summary statistics snakemake pipeline (https://github.com/EBISPOT/gwas-sumstats-harmoniser). Alleles with a minor allele count <10 were filtered out; where minor allele counts were not available, these were imputed from GnomAD v2.1.1[88] European frequencies lifted over to GRCh38. We then filtered all summary statistics to those variants also present at minor allele frequency >0.01 in 1000 genomes phase 3 (unrelated European participants) called against GRCh38[86]. To find independently significant lead loci, we used the Open Targets genetics finemapping pipeline (https://github.com/opentargets/genetics-finemapping) to filter summary statistics to variants with $P < 5 \times 10^{-8}$ (excluding MHC region chr6:28510120-33480577) and performed distance-based clumping of significant variants with a clumping distance of ±500 kb. The number of lead variants identified for each disorder is shown in Supplementary Table 1. ASD was excluded from downstream CHEERS analysis as only two significant loci were detected.

### MDD-schizophrenia discordant T cell acetylation peaks

For the Blueprint and Soskic datasets, MDD-schizophrenia discordant T cell acetylation peaks were defined as those T cell-specific peaks of histone acetylation that were co-located with cis-diagnostic variants defined by the GWAS meta-analysis of MDD but not schizophrenia (or by the GWAS meta-analysis of schizophrenia but not MDD). In each dataset, T cell-specific peaks were consistently defined as those peaks with CHEERS specificity rank >0.9 for any T cell subset. To investigate the properties of these discordant H3K27ac marks, we annotated them by the type of regulatory element (promoter, genic enhancer, or non-genic enhancer), as defined by the Roadmap dataset. For the analysis of discordant T cell peaks in unstimulated cells from the Blueprint dataset, we used regulatory elements defined in unstimulated peripheral blood T cells from the Roadmap dataset; for the analysis of discordant T cell peaks in stimulated CD4+ T cells from the Soskic dataset, we used regulatory elements defined in PMA-stimulated CD4+ T cells in the Roadmap dataset. Overlaps were counted as co-location of one or more base pairs of the H3K27ac peak with the relevant regulatory element.

We also examined whether the MDD-schizophrenia discordant T cell histone acetylation peaks were truly discordant between disorders, or whether their apparent discordance simply reflected thresholding of association statistics, and hence loss of information about subgenome-wide significant associations overlapping the

acetylation peak in the non-implicated disorder. To investigate this further, at each site of co-location of MDD or schizophrenia risk variants with T cell acetylation peaks ("variant-peak overlaps"), we compared the GWAS association statistics for MDD and schizophrenia.

## Over-representation analysis

Following CHEERS analysis, to test which biological pathways were implicated in T cells, we identified those T cell-specific peaks overlapped by disease risk variants, selected the genes overlapping those peaks or with transcription start sites nearest to those peaks, then performed pathway analysis on those genes. More specifically, to define T cell specific peaks, we selected (for a given disorder) the union of peaks highly specific (CHEERS specificity rank >0.9) to any T cell subset in the Soskic immune stimulation dataset which were also overlapped by risk variants for that disorder. For each peak, we used the ChIPseeker seq2gene function[89] to identify the union of those genes overlapping the peak and those genes with a promoter region overlapping the peak, or (if no promoter overlapped the peak) the gene with the nearest transcription start site (up to a maximum of 10 kilobases away). The selected genes were tested for enrichment of GO biological processes and Reactome pathways using a one-sided hypergeometric test via the clusterProfiler enricher function, with Benjamini-Hochberg correction for multiple testing[90].

## Reporting summary

Further information on research design is available in the Nature Research Reporting Summary linked to this article.

## Data availability

All datasets used for this analysis are publicly available (see Supplementary Table 2). The partitioned LD scores for active regulatory elements in Roadmap tissues generated in this study have been deposited at Zenodo under accession code 5153661[91].

## Code availability

All code used for this analysis is publicly available. Custom code is provided at https://github.com/maryellenlynall/psychimmgen2021 and archived at Zenodo under accession code 7125660 [https://doi.org/10.5281/zenodo.7125661][92]. For other code used see Supplementary Table 2.

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

## Acknowledgements

This work was funded by a Medical Research Council award MR/S006257/1 (M.E.L.); NIHR Senior Investigator award (E.T.B); Open Targets grant OTAR040 (B.S., G.T.). Wellcome Trust grant WT206194 (G.T.); US Veterans Affairs Career Development Award (D.F.L.); NIHR Research Professorship RP-2017-08-ST2-002 (M.R.C.). This work was additionally supported by the NIHR Cambridge Biomedical Research Centre. The authors would also like to acknowledge the crucial input of the Sanger Institute Cellular Genetics Informatics team.

## Author contributions

M.E.L. performed all analyses. M.E.L. and E.T.B. conceived the study and wrote the paper. G.T., M.R.C. and E.T.B. provided supervision. B.S., J.H., J.S., D.F.L., G.A.P., R.P., J.G., M.B.S., G.T. and M.R.C. contributed expertise and reviewed and edited the manuscript.

## Competing interests

E.T.B. serves as a member of the scientific advisory boards of Sosei Heptares, Boehringer Ingelheim, Monument Therapeutics, and the Brain & Behavior Research Foundation, and as a consultant for GlaxoSmithKline; he is also a Deputy Editor of *Biological Psychiatry*. J.G. and R.P. are paid for their editorial work for Complex Psychiatry journal. M.B.S. in the past 3 years has received consulting income from Actelion, Acadia Pharmaceuticals, Aptinyx, atai Life Sciences, Boehringer Ingelheim, Bionomics, BioXcel Therapeutics, Clexio, EmpowerPharm, Engrail Therapeutics, GW Pharmaceuticals, Janssen, Jazz Pharmaceuticals, and Roche/Genentech. M.B.S. has stock options in Oxeia Biopharmaceuticals and EpiVario. He is paid for his editorial work on *Depression and Anxiety* (Editor-in-Chief), *Biological Psychiatry* (Deputy Editor), and *UpToDate* (Co-Editor-in-Chief for Psychiatry). He has also received research support from NIH, Department of Veterans Affairs, and the Department of Defense. He is on the scientific advisory board for the Brain and Behavior Research Foundation and the Anxiety and Depression Association of America. The remaining authors declare no competing interests.
