## [Peer Review File · Nature Communications]

Genetic variants associated with psychiatric disorders are enriched at epigenetically active sites in lymphoid cellsREVIEWER COMMENTS

Reviewer #1 (Remarks to the Author):

In this manuscript, the authors explore whether genetic variants associated with risk for neuropsychiatric are enriched within regulatory elements, relevant in immune cells. The authors find, that variants associated with cross-disorder psychiatric risk as well as those associated with schizophrenia and depression are enriched in active regulatory elements of brain as well as independently of lymphoid cells (mainly CD4+) but not myeloid cells. In addition, they identify that while enrichments for MDD and schizophrenia loci are enriched in active enhancer for the same cell type, the specific regulatory elements did not overlap. This work is an important step forward in our understanding of the inter-relatedness of the immune system and psychiatric disorders.

Strengths:

The manuscript is very clearly written and the analyses are well described. The main finding (overlap in lymphoid but not myeloid cells) is replicated using 3 independent datasets (Roadmap, Blueprint, Soskic) as well as using 2 different statistical methods, LD score regression and a direct enrichment of genetic risk variants with cell-specific epigenetic peaks using the CHEERS algorithm. The three data set used are complementary, as each add an additional layer of information.

Minor comments:

I only have minor comments.

The analysis conditioning the LDSC regression by regulatory elements in brain is very interesting, here it would be of interest to put more information on this in the main text, for example include parts of figure S2 into figure1. Could the authors also provide more information on the discordant sites such as type of regulatory elements, differences in mapped genes etc... For the analysis in brain conditioning for fetal brain regulatory elements, it is not clear why this is run separately with the male and female fetal dataset and not both together.

The finding that both schizophrenia and depression showing strong lymphoid enrichment and enrichment in memory CD4 cells, but that the specific peaks overlapped by risk variants for these disorders were not generally shared between them, is intriguing. Figure S7 give a nice illustration, but could the authors expand more on possible differences in type of regulatory elements (using more fine grained annotations maybe). The pathway enrichment analysis presented in figure S8 could be elaborated on more for their mechanistic/functional relevance.

Reviewer #2 (Remarks to the Author):

In general, I found this to be a well-written, well planned out analysis that integrated a number of publicly available datasets with established bioinformatics methods to test the hypothesis that genetic risk factors for psychiatric disorders act, in part, via immune cell types. The results are very compelling and presented in some detail. There are a lot of analyses and it can be challenging to present this number of statistics in an easy to follow manner. The article does require a fair amount of concentration to keep on top of the results, but the narrative does a good job in this regard of stating the question each analysis is addressing. I particularly enjoyed the nuanced insight that the authors provided which is often lacking in other similar analyses. I outline some queries below.

The schizophrenia GWAS is not the most recent, why did they not use (Pardiñas AF, et al. Common

schizophrenia alleles are enriched in mutation-intolerant genes and in regions under strong background selection. Nat Genet. 2018 Mar;50(3):381-389. doi: 10.1038/s41588-018-0059-2. Epub 2018 Feb 26.)?

Is the replication across different epigenetic datasets because the regulatory regions are consistent between them?

The conclusion "the active regulatory elements overlapped by these cis-diagnostic variants (sequence variants associated with schizophrenia or MDD) were specific to each disorder". How confident are the authors in this statement? Have they checked whether the variants are genuinely specific (look at p-value) and just not genome-wide significant? In addition given the size of regulatory elements are much smaller than LD blocks, it is not the case that subtly different tag SNPs are not annotated to the same element.

Can the authors provide some critic of their methods/datasets?

Table S1 is missing the sample size for Autism.

Specify the exact co-ordinates of MHC region that was excluded.

We thank the editor for the opportunity to address the reviewers' comments. In each case, the reviewer's comment is shown in black text, our response is summarised in blue text, and relevant changes to the main text, figures, and supplementary information are excerpted verbatim in blue *italic* text.

Reviewer #1 (Remarks to the Author):

In this manuscript, the authors explore whether genetic variants associated with risk for neuropsychiatric are enriched within regulatory elements, relevant in immune cells. The authors find, that variants associated with cross-disorder psychiatric risk as well as those associated with schizophrenia and depression are enriched in active regulatory elements of brain as well as independently of lymphoid cells (mainly CD4+) but not myeloid cells. In addition, they identify that while enrichments for MDD and schizophrenia loci are enriched in active enhancer for the same cell type, the specific regulatory elements did not overlap. This work is an important step forward in our understanding of the inter-relatedness of the immune system and psychiatric disorders.

Strengths:

The manuscript is very clearly written and the analyses are well described.

The main finding (overlap in lymphoid but not myeloid cells) is replicated using 3 independent datasets (Roadmap, Blueprint, Soskic) as well as using 2 different statistical methods, LD score regression and a direct enrichment of genetic risk variants with cell-specific epigenetic peaks using the CHEERS algorithm. The three data set used are complementary, as each add an additional layer of information.

We thank the reviewer for their positive comments on the insights offered by this work, the clarity and general interest of the manuscript, and the robustness of the findings.

I only have minor comments.

Ref 1/1: The analysis conditioning the LDSC regression by regulatory elements in brain is very interesting, here it would be of interest to put more information on this in the main text, for example include parts of figure S2 into figure1.

We agree it would be helpful to present more of the brain-conditional LDSC analysis in the main text. We have added material from **Supplementary Figure 2** to create the updated **Figure 1** and legend (see below). Thus, the revised version of **Figure 1** now shows the original LDSC analyses, the LDSC analyses adjusted for male and female fetal brain regional annotations, and the LDSC analyses for immune tissues adjusted for all 10 significant brain regions. The full results (across all tissues) for the LDSC analyses adjusted for all 10 brain regions are presented in **Supplementary Figure 2**, as previously.

Figure 1 Trans-diagnostic risk enrichment at epigenetically active sites in brain tissue and, independently, in T cells. (a) Enrichment of trans-diagnostic risk at active regulatory elements in 88 tissues from the Roadmap epigenomics consortium. *P*-values estimated by stratified linkage disequilibrium score regression (LDSC) analysis (see **Methods**) were used to test the null hypotheses that risk variants were not co-located with epigenetically activated sites more frequently than expected by chance, using the false discovery rate ($FDR < 0.05$; orange) to correct for multiple tests across $N=88$ tissues. Tissues with nominally significant enrichment ($P < 0.05$,

blue) are also shown for context. For results in all other tissues see **Supplementary Figure 1**. PFC, prefrontal cortex; HSC, hematopoietic stem cells. (b) Validation of brain-conditioned LDSC modelling. As expected, when the LDSC model for enrichment of adult brain tissues was conditioned on the active regulatory annotations for fetal brain tissue (male and female), there was significant reduction in enrichment across all adult brain tissues (asterisks indicate two-sample Z-tests with $P < 0.05$). (c) Brain-conditioned analysis of enrichment of trans-diagnostic risk variants at active regulatory annotations in immune tissues. Probability of enrichment (log P scale) was estimated by both unconditioned LDSC modelling (left panel of bar chart; same data as in **Figure 1a** but on a different x-axis range of log probabilities); and brain-conditioned LDSC modelling (right panel of bar chart). Conditioning enrichment of immune cells on active regulatory annotations in all brain tissues did not significantly reduce enrichment for any immune tissue (all two-sample Z-tests had $P > 0.05$); but some T cell subsets were no longer significantly enriched at $FDR = 5\%$; see **Supplementary Figure 2** for comparable results in all other tissues. (d) Enrichment of trans-diagnostic risk in enhancers, genic enhancers and active promoters in all immune subsets. Large tiles show results significant at $FDR < 0.05$, to correct for the 78 annotations tested; mid-sized tiles show results significant at $P < 0.05$. Tile fill indicates the P -value rank within each annotation across cell types. There was enrichment of trans-risk at both enhancers and promoters in multiple adaptive immune cell subsets.

Ref 1/2: Could the authors also provide more information on the discordant sites such as type of regulatory elements, differences in mapped genes etc...

Please see our response to the related point **Ref 1/4**.

Ref 1/3: For the analysis in brain conditioning for fetal brain regulatory elements, it is not clear why this is run separately with the male and female fetal dataset and not both together.

We have clarified the rationale for this analysis by adding the following to the main text:

In Methods:

*“We ran the analyses for male and female fetal brain separately as these sex-stratified datasets were provided separately by the Roadmap Epigenomics Consortium (Roadmap Epigenomics Consortium et al. 2015 Nature) and our analysis of sex-stratified data was compatible with prior multi-tissue analyses of Roadmap data (Finucane et al. Nat Genetics 2018, Zhang and Hardison, Nucleic Acids Research 2017). All analyses were thus conducted and reported using tissue classes, sex-stratified for some but not all tissues, as defined by Roadmap (the full list is shown in **Supplementary Figure 1**).”*

In Discussion:

“We also know that many psychiatric conditions show sex differences in prevalence, and we presented enrichment results separately for male and female fetal brains; but most tissue classes included in the multi-tissue epigenetic datasets we analysed were not represented by sex-stratified data. In particular, appropriate sex-stratified epigenetic data on immune cells were not openly available, although such data would likely be informative in further analysis of risks for neuropsychiatric disorders.”

Ref 1/4: The finding that both schizophrenia and depression showing strong lymphoid enrichment and enrichment in memory CD4 cells, but that the specific peaks overlapped by risk variants for these disorders were not generally shared between them, is intriguing. Figure S7 give a nice illustration, but could the authors expand more on possible differences in type of regulatory elements (using more fine grained annotations maybe).

We agree with the reviewer that this is an interesting question and, in the revised paper, we have further expanded on the disorder-specific sites of epigenetic activation of MDD and schizophrenia cis-diagnostic variants in T cells, as follows:

- (a) We compared the GWAS *P*-values of variants that were co-located with T cell acetylation peaks in one disorder with the GWAS *P*-values of the same set of variants for the other disorder. This demonstrated that the T cell acetylation peaks implicated in one disorder were generally not strongly associated with risk for the other disorder.
- (b) We compared the types of regulatory elements co-located with T cell acetylation peaks overlapped by cis-diagnostic risk variants. We did not find substantial differences between disorders in the types of regulatory elements co-located with MDD-specific or schizophrenia-specific acetylation peaks in T cells, or in the distance of these acetylation peaks to the nearest transcriptional start site.
- (c) In terms of the genes nearest to cis-diagnostic variants co-located with T cell acetylation peaks, we found that very few genes (5 in total) were implicated by the analysis of both MDD and schizophrenia.

Added to Methods:

“MDD-schizophrenia discordant T cell acetylation peaks

For the Blueprint and Soskic datasets, MDD-schizophrenia discordant T cell acetylation peaks were defined as those T cell-specific peaks of histone acetylation that were co-located with cis-diagnostic variants defined by the GWAS meta-analysis of MDD but not schizophrenia (or by the GWAS meta-analysis of schizophrenia but not MDD). In each dataset, T cell-specific peaks were consistently defined as those peaks with CHEERS specificity rank >0.9 for any T cell subset. To investigate the properties of these discordant H3K27ac marks, we annotated them by the type of regulatory element (promoter, genic enhancer, or non-genic enhancer), as defined by the Roadmap dataset. For the analysis of discordant T cell peaks in unstimulated cells from the Blueprint dataset, we used regulatory elements defined in unstimulated peripheral blood T cells from the Roadmap dataset; for the analysis of discordant T cell peaks in stimulated CD4⁺ T cells from the Soskic dataset, we used regulatory elements defined in PMA-stimulated CD4⁺ T cells in the Roadmap dataset. Overlaps were counted as co-location of one or more base pairs of the H3K27ac peak with the relevant regulatory element.

We also examined whether the MDD-schizophrenia discordant T cell histone acetylation peaks were truly discordant between disorders, or whether their apparent discordance simply reflected thresholding of association statistics, and hence loss of information about subgenome-wide significant associations overlapping the acetylation peak in the non-implicated disorder. To investigate this further, at each site of co-location of MDD or schizophrenia risk variants with T cell acetylation peaks (‘variant-peak overlaps’), we compared the GWAS association statistics for MDD and schizophrenia.”

Added to results:

“Enrichment of risk for MDD and schizophrenia at active regulatory elements in T cells shows convergence at the cellular scale, but with limited convergence at the molecular scale

*For both the Blueprint and Soskic datasets, both MDD and schizophrenia risk variants were enriched in T cells at H3K27ac histone acetylation marks. There were no significant differences between disorders in terms of the classes of regulatory elements involved (the proportions of promoters, genic enhancers and non-genic enhancers at the implicated peaks), or in terms of the genomic distance between each implicated acetylation peak and the nearest transcriptional start site (see **Supplementary Table 5**). However, this convergence between disorders at a cellular scale was not representative of convergence at the molecular scale of the acetylation peaks overlapped by cis-diagnostic risk variants, which were not generally shared between disorders.*

*For example, there were only two (of 211 total) T cell acetylation peaks implicated in common between MDD and schizophrenia in the Blueprint dataset, and three (of 214 total) in the Soskic dataset. The genes implicated by cis-diagnostic variant-peak overlap in T cells were also largely discordant between disorders. Only 5 genes were consistently implicated in both MDD and schizophrenia: in both the Blueprint and Soskic datasets, COA8/APOPT1 (a proapoptotic mitochondrial protein) and MAD1L1 (a checkpoint protein); and in the Soskic dataset only, FURIN (a protease), SNORD18 (a non-coding RNA), and RP11-73M18.2 (kinesin light chain). We also compared the cis-diagnostic GWAS statistics independently estimated for MDD and schizophrenia at each variant that was co-located with MDD-schizophrenia discordant acetylation peaks (peaks implicated in one but not both disorders). This analysis demonstrated that discordance of acetylation peaks was not simply reflective of sub-genome wide significance of association signals in the disorder where variant-peak overlap was not detected. In fact, as shown in **Supplementary Figure 9**, many of the variants that were co-located with discordant acetylation peaks had different signs (negative vs. positive) or strengths of association with the two disorders.”*

Updated discussion:

“Epigenetically activated sites in T cells, especially cytokine-stimulated CD4⁺ T cells, were most consistently and significantly enriched for cis-diagnostic variants associated with either schizophrenia or MDD. However, at a molecular scale, the active regulatory elements co-located with these cis-diagnostic variants were largely specific to each disorder. Many of the variants driving T cell enrichment in one disorder were not associated with the other disorder, even at a nominal significance threshold. This suggests convergence of risk for schizophrenia and depression at a cellular level in the immune system, i.e., activated T cells, and raises questions about how epigenetic activation at disorder-specific risk variants might relate to the different clinical phenotypes or pathogenic pathways of schizophrenia and depression.”

New Figure

Supplementary Figure 9 Disease association statistics for variants overlapping T cell specific H3K27ac peaks in the Blueprint and Soskic datasets Scatter plots show the cis-diagnostic GWAS statistics independently estimated for MDD (x-axis) and schizophrenia (y-axis) at each variant that was co-located with MDD-schizophrenia discordant T cell acetylation peaks (peaks implicated in one but not both disorders). Statistics are shown for the T cell H3K27ac peaks implicated in both the Blueprint (left hand side) and Soskic (right hand side) epigenetic datasets. Axes use the log scale for disease association P-values, multiplied by the sign of the beta coefficient, to indicate the direction of effect of the variant on disease risk. Lines show the conventional threshold for genome-wide significance of association for both directions of effect in both disorders ($P = 5 \times 10^{-8}$, dashed lines) and the uncorrected threshold ($P = 0.05$, dotted lines). Many of the variants co-located with discordant T cell acetylation peaks have different signs (negative vs. positive) or strengths of association with the two disorders. For the MDD-schizophrenia discordant histone acetylation peaks, in the Soskic dataset, only 40% of the 649 T cell variant-peak overlaps defined by GWAS $P < 5 \times 10^{-8}$ were significantly associated with both disorders, with the same sign of association, even at the nominal level of $P < 0.05$; and 24% of the variant-peak overlaps in one disorder had an opposite sign of association with the other disorder. In the Blueprint dataset, only 39% of the 337 T cell variant-peak overlaps defined by genome-wide $P < 5 \times 10^{-8}$ were significantly associated with both disorders, with the same sign of association, even at the nominal level of $P < 0.05$; and 23% of the variant-peak overlaps in one disorder had an opposite sign of association with the other disorder. If histone acetylation at risk variants for MDD and schizophrenia involved the same loci, but a specific variant simply did not reach genome-wide significance for one of the disorders, we would expect the datapoints to cluster along the $y=x$ line in this plot.

New Table:

Supplementary Table 5 Disease-specific T cell acetylation peaks annotated by class of regulatory elements The properties of T cell specific H3K27ac peaks overlapped by cis-diagnostic GWAS risk variants for MDD only, or for schizophrenia only, were compared. Upper table shows the overlap of MDD and schizophrenia T cell specific peaks from the Blueprint dataset, i.e., unstimulated T cells, with regulatory elements defined in unstimulated peripheral blood T cells from the Roadmap dataset. Lower table shows the overlap of MDD and schizophrenia T cell specific peaks from the Soskic dataset, i.e., stimulated T cells, with regulatory elements defined in the Roadmap PMA-stimulated T cells. P-values show the results of χ^2 tests (for proportion of peaks overlapping each annotation) or Mann-Whitney U test (for distance to nearest transcription start site, TSS). Separate chi-squared tests are performed for each annotation because although the annotations themselves are exclusive categories, a given peak can overlap more than one annotation. For each dataset, the FDR column indicates P-values corrected for the four χ^2 tests performed for that dataset. There were no significant differences between MDD-only and schizophrenia-only T cell peaks in their overlap with regulatory elements or their distance from the nearest TSS.

T cell acetylation peaks associated with genetic risk for MDD or schizophrenia: Blueprint dataset (i.e., non-stimulated cells)				
	MDD-only peaks	Schizophrenia-only peaks	χ^2 test (P value)	FDR
Overlap with T cell non-genic enhancers	39%	28%	0.07	0.1
Overlap with T cell genic enhancers	38%	25%	0.04	0.1
Overlap with T cell promoters	22%	13%	0.1	0.2
Distance to nearest TSS (median)	14892 bp	14747 bp	0.6	0.6
T cell acetylation peaks associated with genetic risk for MDD or schizophrenia: Soskic dataset (i.e., stimulated cells)				
	MDD-only peaks	Schizophrenia-only peaks	χ^2 test (P value)	FDR
Overlap with stimulated T cell non-genic enhancers	51%	52%	0.7	0.9
Overlap with stimulated T cell genic enhancers	48%	47%	0.5	0.9
Overlap with stimulated T cell promoters	33%	33%	0.9	0.9
Distance to nearest TSS (median)	11141 bp	9258 bp	0.5	0.9

Ref 1/5: The pathway enrichment analysis presented in figure S8 could be elaborated on more for their mechanistic/functional relevance.

We agree that further detail on the enrichment analysis would be helpful to readers. We have added the names of the genes driving the pathway enrichment in T cells to the updated **Supplementary Figure 8** (see below), and provided the disorder-specific lists of nearest genes as a new **Supplementary Table 6**. We have also updated the results section to provide further detail on the nearest genes results - see response to **Ref 1/4**, and the additional text below:

Added to Results:

“Trans-risk and cis-risk for schizophrenia showed enrichment of pathways including epigenetic regulation, pre-notch processing, and estrogen-dependent gene expression in T cells, in large part driven by histones and histone-related genes (Figure 4D and Supplementary Figure 8). Cis-risk for depression showed enrichment in negative regulation of cold-induced thermogenesis and in dendrite development in T cells. In contrast, rheumatoid arthritis showed enrichment of lymphoid cell differentiation, activation, and response to antigenic stimulus (Supplementary Figure 8). Notably, most of the T cell genes highlighted by the epigenetic analysis of trans-risk, or cis-risks for schizophrenia and depression (see Supplementary Table 6), did not feature in any enriched

pathways, perhaps because the immunobiology relevant to psychiatric disorders has not yet been captured in pathway databases.”

Updated Figure

Supplementary Figure 8 Pathway enrichment for genes nearest to T cell acetylation peaks co-located with trans-diagnostic and cis-diagnostic variants. Plots show the most significantly enriched pathways in T cells for each disorder or group of disorders, based on the genes nearest to the T cell-specific H3K27ac peaks co-located with risk variants. X-axes show log scale P values for enrichment of the pathways; circle fill shows gene ratio (number of nearest genes in the pathway / total number of test genes); listed genes are those genes in the enriched pathway nearest to the T cell peaks. Plots are shown only for those disorders which showed enrichment of risk variants in T cell subsets. Pathway over-representation analysis was performed using Reactome and GO Biological Process pathways. Only pathways with FDR < 0.05 are shown, with a maximum of 10 pathways shown per disorder or group of disorders.

Reviewer #2 (Remarks to the Author):

In general, I found this to be a well-written, well planned out analysis that integrated a number of publicly available datasets with established bioinformatics methods to test the hypothesis that genetic risk factors for psychiatric disorders act, in part, via immune cell types. The results are very compelling and presented in some detail. There are a lot of analyses and it can be challenging to present this number of statistics in an easy to follow manner. The article does require a fair amount of concentration to keep on top of the results, but the narrative does a good job in this regard of stating the question each analysis is addressing. I particularly enjoyed the nuanced insight that the authors provided which is often lacking in other similar analyses.

We thank the reviewer for their positive comments on the strength of our results and our approach to their interpretation.

I outline some queries below.

Ref 2/1 The schizophrenia GWAS is not the most recent, why did they not use (Pardiñas AF, et al. Common schizophrenia alleles are enriched in mutation-intolerant genes and in regions under strong background selection. *Nat Genet.* 2018 Mar;50(3):381-389. doi: 10.1038/s41588-018-0059-2. Epub 2018 Feb 26.)?

We agree with the reviewer that the Pardiñas dataset would have been a very reasonable alternative choice. Since 2018, further relevant datasets with larger sample sizes have also become available (e.g., Trubetskoy *et al.* *Nature* 2022 <https://doi.org/10.1038/s41586-022-04434-5>). The results presented in this paper related to schizophrenia are predicated on a large well-powered GWAS meta-analysis with considerable overlap with the Pardiñas meta-analysis. We have moderated the language in the paper to reflect this as follows:

Added to discussion:

“The genetic architecture of psychiatric disorders is currently incomplete. As more novel risk variants are identified in future, the number of epigenetically active loci implicated in adaptive immune cells will likely increase, and understanding of their functional implications will be further refined. However, analyses based on alternative European GWAS datasets are unlikely to alter our major findings, which focus mainly on patterns of enrichment across different cellular subsets for risk variants that have already been significantly associated with one or more psychiatric disorders. It seems unlikely that further expansion of data available for GWAS in future will result in currently significant variants becoming less significantly associated with risk of psychiatric disorder(s). In short, we expect our current results to represent a robust core set of acetylated regions in T cells which will be enhanced rather than undermined by future increase in the scale and dimensionality of GWAS studies in psychiatry.”

Ref 2/2 Is the replication across different epigenetic datasets because the regulatory regions are consistent between them?

This is a very interesting question. On further analysis, we found that the enrichment of risk variants at T cell-specific acetylation peaks is not simply due to shared regulatory regions across the datasets. The two most similar datasets are the Blueprint and Soskic datasets, as both are based on H3K27ac marks in sorted cell subsets, but many of the peaks driving the T cell enrichment in the two datasets are different. We have added the following to the manuscript:

Added to Methods

“The three epigenetic datasets we used have some overlap in the immune cell subsets represented. For example, CD4⁺ T cells were represented in all three datasets; monocytes and B

cells were represented in both the Roadmap and Blueprint datasets; and macrophages were represented in both the Roadmap and Soskic datasets. However, the annotations used for the Roadmap analysis (“active states”, comprising enhancers and promoters defined using multiple histone marks) are different from the annotations used for the Blueprint and Soskic analyses (cell subtype-specific H3K27ac marks).”

Added to Results:

“We note that many of the T-cell specific histone acetylation peaks in the Blueprint data which were co-located with risk variants were not also overlapped by T-cell specific peaks in the Soskic dataset co-located with risk variants (55% for MDD and 54% for schizophrenia). Likewise, many of the Soskic T-cell peaks co-located with risk variants were not also overlapped by any of the Blueprint T-cell peaks co-located with risk variants (36% for MDD and 29% for schizophrenia). This suggests that the replicable T cell enrichment observed was not driven exclusively by similarities between the specific peaks detected in the different datasets.”

Ref 2/3 The conclusion “the active regulatory elements overlapped by these cis-diagnostic variants (sequence variants associated with schizophrenia or MDD) were specific to each disorder”. How confident are the authors in this statement? Have they checked whether the variants are genuinely specific (look at p-value) and just not genome-wide significant?

We agree that this is an interesting question and we have further expanded on the disorder-specific sites of epigenetic activation of MDD and schizophrenia cis-diagnostic variants in T cells. We compared the GWAS *P*-values of variants that were co-located with T cell acetylation peaks in one disorder with the *P*-values of the same set of variants estimated by GWAS for the other disorder. This demonstrated that the T cell acetylation peaks implicated in one disorder were generally not strongly associated with risk for the other disorder. For more detail, see the response to reviewer comment **Ref 1/4** above, with new text added to the manuscript and a new **Supplementary Figure 9** which addresses this important point.

Ref 2/4 In addition given the size of regulatory elements are much smaller than LD blocks, it is not the case that subtly different tag SNPs are not annotated to the same element.

The method we used (CHEERS) is not sensitive to subtle differences in tag variants, as CHEERS is sensitive to overlap of any variants in the lead variant’s LD block with the regulatory element. We agree that this is an important point and we have updated the text to clarify this as follows.

Added to Methods:

“In brief, CHEERS identifies histone acetylation peaks (or other genomic annotations) which overlap lead variants or variants in strong LD ($r^2 > 0.8$) with lead variants; then calculates the mean cell type specificity score (in that cell type) of those peaks, which captures the degree of enrichment of that cell type for a given disorder. Seeking overlap between regulatory elements and any variant in the LD block of a given lead variant ensures that the CHEERS method is not sensitive to subtle differences in tag variants between different association studies.”

Ref 2/5 Can the authors provide some critic of their methods/datasets?

Of course, our methods have their limitations, which we have already addressed somewhat in the the Discussion (lines 538-573), where we cover limitations stemming from population ancestry, the availability of epigenetic datasets, and difficulty interpreting the strength of immune enrichment in the face of potentially heterogeneous within-disorder pathophysiology. However, we have added further additional self-critical remarks at the reviewer’s request, including comments on sex

differences (see **Ref 1/3**) and the incomplete genetic architecture of psychiatric disorders (see **Ref 2/1**). New text as follows:

Added to discussion:

The genetic architecture of psychiatric disorders is currently incomplete. As more novel risk variants are identified in future, the number of epigenetically active loci implicated in adaptive immune cells will likely increase, and understanding of their functional implications will be further refined.

...

“We also know that many psychiatric conditions show sex differences in prevalence, and we presented enrichment results separately for male and female fetal brains; but most tissue classes included in the multi-tissue epigenetic datasets we analysed were not represented by sex-stratified data. In particular, appropriate sex-stratified epigenetic data on immune cells were not openly available, although such data would likely be informative in further analysis of risks for neuropsychiatric disorders.”

Ref 2/6 Table S1 is missing the sample size for Autism.

We thank the reviewer for their careful review and for spotting this important omission. We have added the sample size to **Supplementary Table 1**, as suggested.

Excerpt from updated Table S1

Study	Number cases	Number controls	Number of genome-wide independently significant loci	Download link
Cross-disorder psychiatric risk ¹	162,151	276,846	115	https://pgcdata.med.unc.edu/cross_disorder/pgc_cdg2_meta_no23andMe_oct2019_v2.txt.daner.txt.gz
Depression ²	264,984	581,929	122	dbGaP Study Accession: phs001672.v6.p1
Schizophrenia ³	36,989	113,075	108	https://pgcdata.med.unc.edu/schizophrenia/ckqny.scz2snpres.gz
Bipolar disorder ⁴	20,352	31,358	16	https://www.med.unc.edu/pgc/download-results/ File = daner_PGC_BIP32b_mds7a_0416a
Autism ⁵	18,382	27,969	2	https://pgcdata.med.unc.edu/autism_spectrum_disorders/iPSYCH-PGC_ASD_Nov2017.gz

Ref 2/7 Specify the exact co-ordinates of MHC region that was excluded.

We excluded the same MHC region as in the published method for LDSC (Finucane *et al.* 2015 *Nature Genetics*) and have clarified this as follows:

Added to Methods:

“Summary statistics were preprocessed using the LDSC recommended script `munge_sumstats.py` and we performed LDSC for each tissue in the Roadmap dataset, using recommended settings, excluding the extended MHC region (GRCh37 chr6:25-34 Mb).”

REVIEWERS' COMMENTS

Reviewer #1 (Remarks to the Author):

The authors have been extremely responsive to this reviewer's comments and the added methodological details, figures and results have further improved the quality of the manuscript. I recommend acceptance for publication.

Reviewer #2 (Remarks to the Author):

The authors have addressed all my concerns. I might suggest that the authors reconsider this statement:

"It seems unlikely that further expansion of data available for GWAS in future will result in currently significant variants becoming less significantly associated with risk of psychiatric disorder(s)"

While this makes statistical sense, it does happen. Variants significance in one GWAS, become insignificant in the next meta-analysis. It doesn't mean they are false positives, but I might think about the phrasing here. I don't think this level of explanation is needed in the manuscript.

We thank the editor for the opportunity to address the reviewers' comments. In each case, the reviewer's comment is shown in black text, our response is summarised in blue text, and relevant changes to the text are excerpted verbatim in blue *italic* text.

Reviewer #1 (Remarks to the Author):

The authors have been extremely responsive to this reviewer's comments and the added methodological details, figures and results have further improved the quality of the manuscript. I recommend acceptance for publication.

We are pleased that our updated manuscript has addressed the reviewer's initial comments and queries. We would like to thank the reviewer for their very helpful input.

Reviewer #2 (Remarks to the Author):

The authors have addressed all my concerns.

We thank the reviewer for their very helpful comments and input to the manuscript.

Ref 2/1 I might suggest that the authors reconsider this statement:

"It seems unlikely that further expansion of data available for GWAS in future will result in currently significant variants becoming less significantly associated with risk of psychiatric disorder(s)"

While this makes statistical sense, it does happen. Variants significance in one GWAS, become insignificant in the next meta-analysis. It doesn't mean they are false positives, but I might think about the phrasing here. I don't think this level of explanation is needed in the manuscript.

We thank the reviewer for this point and agree that this sentence could be misleading. As commented by the reviewer, this discussion is not central to the manuscript, so we have simply removed the problematic sentence from the discussion. The updated section of the discussion now reads as follows:

Updated discussion:

"The genetic architecture of psychiatric disorders is currently incomplete. As more novel risk variants are identified in future, the number of epigenetically active loci implicated in adaptive immune cells will likely increase, and understanding of their functional implications will be further refined. However, analyses based on alternative European GWAS datasets are unlikely to alter our major findings, which focus mainly on patterns of enrichment across different cellular subsets for risk variants that have already been significantly associated with one or more psychiatric disorders. In short, we expect our current results to represent a robust core set of acetylated regions in T cells which will be enhanced rather than undermined by future increase in the scale and dimensionality of GWAS studies in psychiatry."